# Use of Plant Extracts, Bee-Derived Products, and Probiotic-Related Applications to Fight Multidrug-Resistant Pathogens in the Post-Antibiotic Era

António Machado [1,*], Lizbeth Zamora-Mendoza [2], Frank Alexis [2,*] and José Miguel Álvarez-Suarez [3,*]

1. Laboratorio de Bacteriología, Colegio de Ciencias Biológicas y Ambientales COCIBA, Instituto de Microbiología, Universidad San Francisco de Quito USFQ, Quito 170901, Ecuador
2. Colegio de Ciencias e Ingenierías, Departamento de Ingeniería Química, Universidad San Francisco de Quito USFQ, Quito 170901, Ecuador; lzamora@estud.usfq.edu.ec
3. Colegio de Ciencias e Ingenierías, Departamento de Ingeniería en Alimentos, Universidad San Francisco de Quito USFQ, Quito 170901, Ecuador
* Correspondence: amachado@usfq.edu.ec (A.M.); falexis@usfq.edu.ec (F.A.); jalvarez@usfq.edu.ec (J.M.Á.-S.)

**Abstract:** The 'post-antibiotic' era is near according to the World Health Organization (WHO). It is well known, due to the work of the scientific community, that drugs (antibiotics, antifungals, and other antimicrobial agents) are continuously becoming less effective, and multidrug-resistant (MDR) pathogens are on the rise. This scenario raises concerns of an impending global infectious disease crisis, wherein a simple opportunistic infection could be deadly for humans. The war against MDR pathogens requires innovation and a multidisciplinary approach. The present study provides comprehensive coverage of relevant topics concerning new antimicrobial drugs; it suggests that a combination of different natural products (such as plant extracts, honey, propolis, prebiotics, probiotics, synbiotics, and postbiotics), together with drug therapy, could be used as an adjuvant in standard treatments, thus allowing drug sensitivity in MDR pathogens to be restored, host immunity to be enhanced, and clinical efficiency to be improved. Currently, new and relevant developments in genomics, transcriptomics, and proteomics are available for research, which could lead to the discovery of new antimicrobial drugs and a new generation of antibiotics and non-antibiotics. However, several areas concerning natural products and their combination with standard drugs remain unclear. In an effort to advance new therapies for humankind, these gaps in the literature need to be addressed.

**Keywords:** multidrug-resistant (MDR) pathogens; natural products; plant extracts; honey; prebiotics; probiotics; synbiotics; postbiotics; effectiveness of clinical treatment

## 1. Introduction

The World Health Organization (WHO) expects the 'post-antibiotic' era to occur around the year 2050 after evaluating data from 129 member states; every region of the world showed extensive resistance to antimicrobial agents [1,2]. The overuse of antibiotics in several different areas, such as agriculture (to promote livestock growth) and in hospitals (to order standard treatments), has quickly led to the proliferation of drug-resistant bacteria being spread via human travel and poor sanitation practices worldwide [1,3–5]. Antimicrobial resistance (AMR) is currently an international issue, and millions of people die every year as a result of opportunistic or primary pathogens that have become resistant due to horizontal gene transfer (HGT) mechanisms and/or biofilm formation [6,7]; this is a multi-faceted problem with a catastrophic impact on everyone, including humans, livestock, and the environment [8]. This has led to the estimation that, in 2050, 10 million people will die of infections that cannot be treated because of resistant bacteria and ineffective antibiotics [9].

Currently, healthcare-associated infections (HAI) comprise a main public health concern. These infections usually occur 48 h after hospitalization, although they may also occur

after patients are discharged [10]. It is estimated that 7% and 10% of hospitalized patients in developing and developed countries are affected by HAI [10,11], respectively. Moreover, around 3.2 million patients per year are affected by HAI in Europe [10]. The mortality rate and incidence among patients are normally correlated with the patients' immunological status and geographical region; however, patients in burn units and intensive care units (ICUs), as well as organ transplant receivers and neonates, are the most common hospital­ized groups affected by HAI [5,12]. In addition, these infections are also responsible for three out of four lethal cases in neonates in Sub-Saharan Africa and South-East Asia [10]. The most reported HAI are surgical site infections (2–5% incidence rate), catheter-related blood stream infections (12–25% incidence rate), catheter-related urinary tract infections (12% incidence rate), and ventilator-associated pneumonia (9–27% incidence rate) [13]. Currently, the most worrisome global AMRs are the plasmid-mediated spread of carbapen­emases (e.g., KPC, NDM, VIM, OXA-48, and OXA-51) and colistin-resistance genes (*mcr*) in *Enterobacteriaceae*, *Acinetobacter baumannii*, and *Pseudomonas aeruginosa*, as well as the vancomycin resistance gene (*vanA*) in *Enterococcus* sp. and *Staphylococcus aureus*, and the methicillin resistance gene (*mecA*) in *S. aureus* [10,13].

This review highlights essential factors contributing to AMR, the epidemiology of the resistant bacteria, and novel alternative therapies that should be developed in subsequent decades to fight the rise of multidrug-resistant (MDR) pathogens. The scientific community and the general public must understand and cooperatively implement the 'One Health ap­proach' [8,14]. Neglecting the AMR problem will anticipate the arrival of the 'post-antibiotic era'; the overuse of antibiotics will increase healthcare costs, morbidity, mortality, and envi­ronmental degradation [15–18]. The lack of new medicines for effective treatments against MDR pathogens, and the emergence of these microorganisms, is a growing global public health concern [19,20]. Despite the critical need for new antimicrobial agents, their rate of development is decreasing [5,12]. Fighting MDR infections calls for a multidisciplinary approach; the present review discusses three alternative antimicrobial drugs and suggests that the combination of different natural products, together with drug therapy, could be used as an adjuvant in standard treatment in order to restore drug sensitivity, enhance host immunity, and improve clinical efficiency. The first section describes the rise of AMR, and the second and third sections discuss natural products with antimicrobial activities, such as plant and honey extracts, respectively. Finally, the fourth section discusses the recent and ongoing developments in microbiome research that are enabling the formulation of new prebiotic, probiotic, and postbiotic products. Therefore, the promising solutions found during the development of new agents are encouraged in the present work. We believe that the success of the long-term battle against MDR pathogens will require new strategies that target other and multiple cellular processes.

## 2. Rise of MDR Pathogens and Future Trends concerning the Global Infectious Disease Crisis

Understanding the molecular mechanisms underlying antimicrobial (antibiotic or anti­fungal) resistance is essential, and it requires a deep knowledge of microbial structures and their metabolic functions. Structural and metabolic differences between microorganisms and host cells make it possible to selectively kill the pathogen, or at least inhibit its growth with antimicrobial agents [21], thus allowing the host immune system to eliminate the infection [7]. AMR in pathogens (particularly in bacteria) has emerged as a global chal­lenge since antibiotics were first administered, as it threatens the effectiveness of clinical treatments. In recent decades, there has been an exponential rise in antibiotic resistance-associated factors in microbial communities, most likely driven by the mobility of virulence genes through HGT mechanisms (such as transformation, conjugation, and transduction). Although conjugation, transformation, and transduction are the three primary processes of HGT, six major types of mobile genetic elements (MGEs) have been characterized in the MDR pathogens, such as transposons, gene cassettes, integrons, genomic islands, plasmids, bacteriophages, and integrative conjugative elements (ICEs) [22]. These HGT mechanisms

induce genome evolution, and it has caused the rise of different and successful MDR pathogens worldwide [23]. Furthermore, the United States National Institutes of Health (NIH) revealed that around 65% and 80% of all microbial and chronic infections are associated with biofilm formation [24]. The process of biofilm formation consists of many steps, starting with attachment to a living or non-living surface; this leads to the formation of a micro-colony, giving rise to three-dimensional structures, and after maturation, detachment occurs [25,26]. During the formation of biofilm, several species communicate with one another as they employ quorum sensing [27]. In general, biofilms show resistance against the human immune system, as well as against disinfectants and antimicrobials (antibiotics and/or antifungals) [5]. In summary, the understanding of microbial biofilm is important to manage and/or to eradicate biofilm-related diseases. It is believed that biofilms have a great impact on the dissemination of antibiotic resistance as they facilitate HGT mechanisms. Due to the high cell density in the biofilm structure, there is a significant increase in HGT mechanisms. Moreover, the protection given by the extracellular polymeric substances (EPS) of the biofilm by itself intensifies the AMR of the infections with or without the presence of resistant genes in the microbial populations within the biofilm (Figure 1).

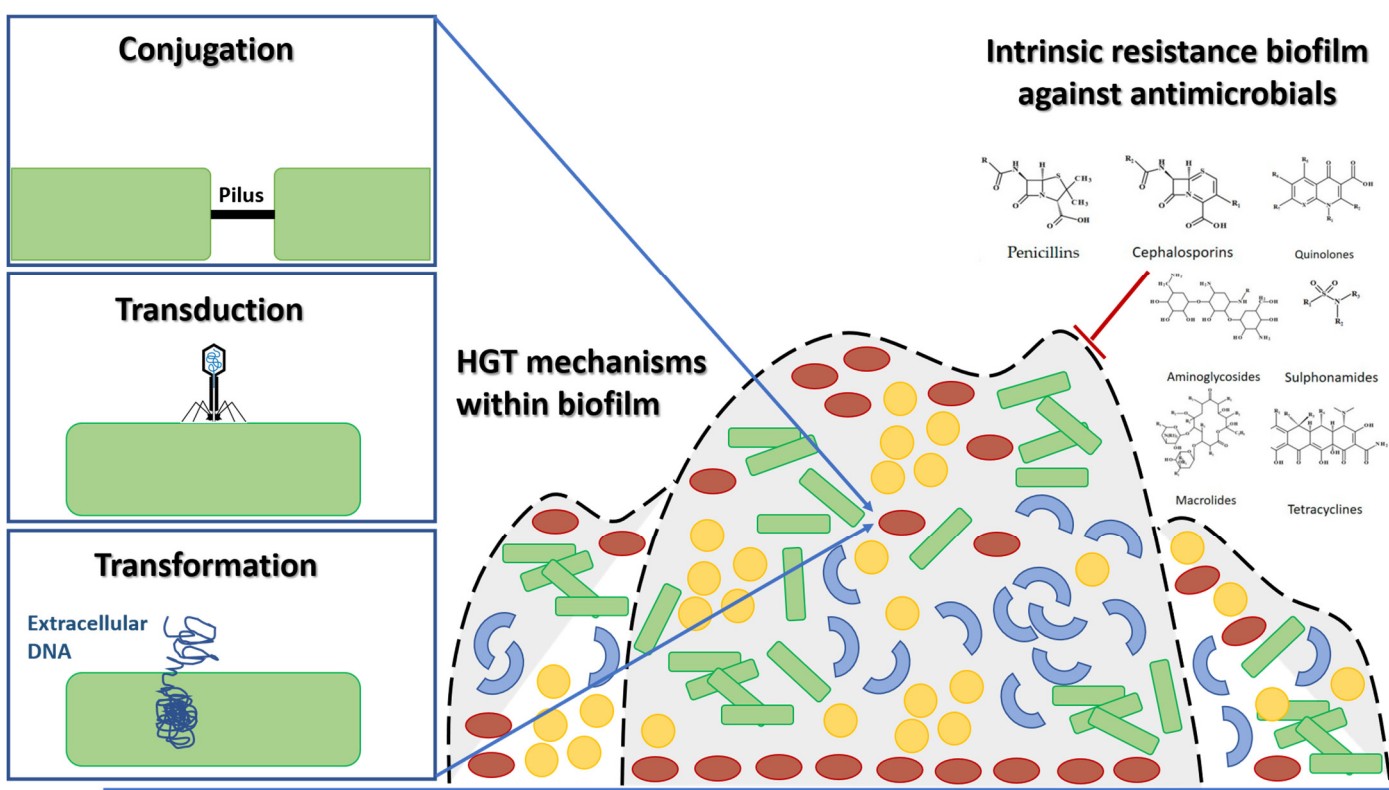

**Figure 1.** Conceptual hypothesis model concerning the rise of HGT mechanisms and the intrinsic resistance of pathogens during biofilm formation, which occurs during infections.

Until recently, microorganisms were typically observed as unicellular organisms in the environment; however, currently, it is well known that they prefer to form multicellular communities in nature. As a result, these so-called biofilms are able to endure harsher environments, such as those with a lack of nutrients, natural competitors, and even toxic elements (such as antibiotics) [6,25]. Most microorganisms can live together in biofilms, and the majority of them are known as polymicrobial biofilms, wherein each species and their cells show distinct features when compared with their planktonic form [28]. More specifically, these features include heterogeneity of gene expression, division of roles in the community, presence of persister cells, and enhanced tolerance to antibiotics [29]. In fact, persister cells are dormant and non-dividing cells that exhibit multidrug tolerance and

survive treatment from all known antimicrobials [29]. In addition, biofilms are embedded with a highly dense matrix of extracellular polymeric substances (EPS). EPS are complex and potentially diverse polymers produced by the cells within biofilms; they are usually composed of exopolysaccharides, amyloid-like proteins, lipids, and extracellular DNA (eDNA) [26,30]. Despite the intrinsic resistance caused by the presence of EPS surrounding the microbial community within the biofilm, there are also several mechanisms of AMR that have been previously described in other reports; these reports focus on the inactivation or modification of a drug, limiting the drug uptake, drug target modifications, and reducing the active concentration of a drug inside a cell via drug efflux [22,31,32]. The physiological adaptation of microorganisms within a biofilm induces the development of intrinsic resistance, and biofilms are a leading example of resistance to antimicrobial products. Biofilms have been reported to be 100–1000 times more resilient against antimicrobials when compared with equivalent planktonic counterparts [31]. In addition to traversing through the passage of the EPS of the biofilm, the antimicrobial drug needs to enter the microorganism cell membrane at an adequate concentration for a substantial time in order to perform its pharmacological action and produce antimicrobial activity. The efflux pump mechanism is a common mechanism that numerous MDR pathogens use to counter drugs; it extrudes antimicrobial agents faster than it would otherwise [32]. Efflux pumps are proteinaceous, and they are membranal transporters that are able to regulate the microbial cell cytoplasmatic environment; therefore, they can remove toxins, antifungals, and antibiotics. Based on their sequence homologies, the source of energy, substrate binding, and structural components of efflux transporters are usually classified into five prominent families [32]. More specifically, these families are as follows: resistance-nodulation division (RND); adenosine triphosphate (ATP)-binding cassette superfamily (ABC); multidrug and toxic compound extrusion (MATE); major facilitator superfamily (MFS); and small multidrug resistance family (SMR). The two main families of efflux pump proteins in fungi are the ABC and MFS transporters, whereas the RND family is a specific group for Gram-negative bacteria. Finally, the ABC, MATE, MFS, and SMR families are found in Gram-positive and Gram-negative bacteria in significant quantities [32]. However, it is also known that different microorganisms can alter their membrane permeability due to the over- or under-expression of porins, thus controlling the passage of several compounds through the cell membrane; this has been widely reported among Gram-negative pathogens [32]. Finally, another well-known AMR mechanism is the degradation of antimicrobial agents or target site modifications using enzymes. In fact, different enzymes can remove or add a specific moiety to the antibiotic molecule or target site, thus causing a successful mutation on the pathogen which acts against a certain drug [22]. The modification of antimicrobial drugs can be achieved via numerous biochemical reactions that are catalyzed by enzymes; they involve phosphorylation (e.g., chloramphenicol and aminoglycosides), acetylation (e.g., chloramphenicol, aminoglycosides, and streptogramins), and adenylation (e.g., lincosamides and aminoglycosides) [22]. Likewise, target site modifications of several drugs can be succeeded by point mutations in the gene encoding target site, causing an enzymatic change to the target site, and it bypasses the original site [22].

MDR pathogens are one of the most important current threats to public health, and typically, these pathogens are associated with nosocomial and biofilm-related infections [5,33]. In addition, the ineffectiveness of available treatments for such infections has been reported in numerous studies [12,20,34]. Although antibiotics have made it possible to treat deadly infections, the overuse and misuse of antibiotics in recent decades have accelerated the spread of AMR, causing treatments to become ineffective. Currently, at least 700,000 people worldwide die each year due to MDR pathogen-related infections [35]. In 2017, due to the increasing number of AMR reports, the WHO published a list of pathogens, including the pathogens that comprise the acronym ESKAPE (*Enterococcus faecium*, *S. aureus*, *Klebsiella pneumoniae*, *A. baumannii*, *P. aeruginosa*, and *Enterobacter species*), which were given the highest "priority status" as they represent the greatest threat to humans [36]. Several reports note that the rise of these MDR, as well as extensively drug-resistant (XDR) pathogens, render even the most

effective drugs ineffective [37,38]. Likewise, in 2022, Cangui and colleagues evaluated the prevalence of biofilms in central venous catheters (CVC) when investigating CVC-related infections in Intensive Care Unit (ICU) patients worldwide. CVC is considered to be one of the deadliest nosocomial or hospital-acquired infections, as *S. aureus*, coagulase-negative staphylococci, *A. baumannii*, and *P. aeruginosa* are the most frequently isolated pathogens [5]. In particular, extended-spectrum β-lactamase (ESBL) and carbapenemase-producing Gram-negative bacteria have emerged as an important therapeutic challenge [4,35,39,40]. Therefore, the development of novel therapies to treat drug-resistant infections, especially those caused by ESKAPE pathogens, is of utmost importance [41,42].

## 3. Plant Extracts

Historically, plant extracts have been used to treat several bacterial infections in medicine [43]. Several techniques have allowed the extraction and identification of bioactive compounds to recognize the mechanism of action that causes bacteriostatic effects [44,45]. Plant extracts which have antibacterial effects can be obtained from roots, fruits, flowers, stems, leaves, and seeds. Plant extracts are mainly composed of two types of metabolites which are classified as primary and secondary compounds. Primary metabolites are essential compounds for plant survival, whereas secondary metabolites are usually formed in response to plant interactions with the environment [32]. Therefore, primary metabolites are usually products of glycolysis, the shikimate pathway, and the tricarboxylic acid cycle, among other functions which are involved in nutrition and reproduction. Moreover, these metabolites can also act as a precursor to thousands of secondary metabolites that are produced at different steps of the primary metabolic pathways, producing new compounds that facilitate plant adaptations against environmental stress (e.g., bacteria, fungi, insects, disease, injury, temperature, and moisture) [32]. The molecules with the greatest antimicrobial effects are usually secondary metabolites; the most common of these molecules identified in most studies are terpenoids, polyphenols (such as flavonoids, stilbenes, lignans, and phenolic acids), and alkaloids. Many plant terpenoids have found fortuitous uses in medicine [46], and their antimicrobial activity has been attributed to their general membrane disrupting properties [46]. For example, there is evidence that terpenoids of *Syzigium cumini* exhibit antibacterial activity against methicillin-resistant *S. aureus* (MRSA) and pathogenic *E. coli* when minimum inhibitory concentration (MIC) and minimum bactericidal concentration (MBC) assays are used [32]. Likewise, polyphenols' antimicrobial effects were also documented [32], as were their antioxidant, anti-inflammatory, anticancer, and antihypertensive activities [32,47]. Although the exact mechanism for polyphenols' antimicrobial action is not fully understood, several polyphenols were reported to exhibit antimicrobial activity against MDR pathogens. Studies postulated that there are different mechanisms at the cellular level, wherein polyphenols can bind to bacterial enzymes via a hydrogen bond, inducing several modifications in terms of cell membrane permeability and cell wall integrity [32,47]. Numerous reports were focused on the most abundant flavonoids, such as flavanols (e.g., quercetin and kaempferol), which demonstrated potent antimicrobial activity against Gram-positive and Gram-negative pathogens, as well as resistant strains [32]. As previously described, the combination of quercetin with amoxicillin exhibited synergistic activity against amoxicillin-resistant *Staphylococcus epidermidis* isolates [47]. *Bryophyllum pinnatum* extract revealed that kaempferol, and its derivatives, exhibited significant antimicrobial activity against several bacterial and fungal pathogens, including antibiotic-resistant *S. aureus* and *P. aeruginosa*, as well as *Candida* species and *Cryptococcus neoformans* [48]. The kaempferol-mediated inhibition of the NorA efflux pump was postulated to be an action mechanism against *S. aureus* [49]. Finally, alkaloids are organic nitrogenous compounds that are structurally diverse, and their antimicrobial activity has been reported since the 1940s [32]. The mechanism of alkaloids which works against various microbial pathogens is characterized by efflux pump inhibition [50]. In 2020, Duda-Madej and colleagues demonstrated the antibacterial activity of 18 compounds of the *O*-alkyl derivatives of naringenin and their oximes; these compounds worked against

clinical isolates of clarithromycin-resistant *Helicobacter pylori*, vancomycin-resistant *Enterococcus faecalis*, methicillin-resistant *S. aureus*, and beta-lactam-resistant *A. baumannii* and *K. pneumoniae*. [51]. Of the pathogen group set, the clarithromycin-resistant strain of *H. pylori* showed the highest susceptibility to most of the 18 compounds. Moreover, when evaluating the synergy between the O-alkyl derivatives/oximes and several antibiotics via the fractional inhibitory concentration index (FICI), the synergy was observed to be potent when used against *H. pylori*, *S. aureus*, and *E. faecalis* [51].

Nonetheless, other types of compounds have been extensively characterized over the last decade. The pure compounds that have been studied most in recent years are andrographolide, borneol, caffeic acid, thymol, citral, quercetin, epigallocatechin gallate, hydroquinone, oridonin, rhodomyrtosone B, and ursolic acid, among others [52]. Depending on the phytochemical compound, the mechanism of action in different bacteria are cell membrane rupture, aerobic metabolism of interference, protein biosynthesis inhibition, DNA segregation, inhibition of respiratory chain complex proteins, damage to cells' structural integrity, and disruption of metabolic pathways [52]. The interest in plant extracts has resulted in different patents over the last 20 years, as described in Table 1.

**Table 1.** Plant extract patents and their applications.

| Title | Date | Patents | Country | Plants | Bacteria | Application | References |
|---|---|---|---|---|---|---|---|
| Antibacterial essential oil | 2023 | CN115708794A | China | Grape, *Zedoariae rhizoma*, *Radix angelicae pubescentis*, myrrh, *Ligusticum wallichii*, *Eucalyptus globulus*, *Boswellia carterii*, clove, peppermint, and coriander | *Streptococcus pyogenes*, *S. aureus*, and *K. pneumoniae* | It inhibits the formation of a biofilm on the surface of a biological material. | [53] |
| An antibacterial and anti-inflammatory composition containing plant extracts that provides itching relief, to be applied accordingly | 2022 | CN115844777A | China | Basil, bergamot, *Salvia miltiorrhiza*, witch hazel, aloe, mint, juniper berry, camellia seed, calendula, *Polygonum multiflorum*, honeysuckle, camphor tree, pseudo-ginseng, honeysuckle, olive, camellia, tea, daphne, *Gentiana rigescens*, *Polygonatum kingianum*, licorice, and *Chrysanthemum* | *Escherichia coli*, *S. aureus*, and *Candida albicans* | It can be used in oral care products, medicines, and skin care products, it provides relief from itching, and it has antibacterial and anti-inflammatory effects. | [54] |
| Plant composite antibacterial agent containing peony extracts, to be prepared and applied accordingly | 2022 | CN115399343B | China | *Scutellaria baicalensis*, aloe, selfheal, honeysuckle, *Pogostemon cablin*, oregano, clove, lavender, and *Folium artemisiae argyi* or *Forsythia suspensa* | *S. aureus*, *C. albicans*, *E. coli*, and *P. Aeruginosa* | It is highly sanitary and safe, and it has low metal corrosion and low skin irritation.It has fast-acting, highly efficient, and long-lasting properties. | [55] |
| Antibacterial hand sanitizer composition containing plant extracts | 2021 | KR102424044B1 | South Korea | Leek, green onion, purslane, water parsley, and perilla leaves | *E. coli*, *S. aureus*, and *S. epidermidis* | It has excellent moisturizing abilities to help maintain skin health. Moreover, it is possible to formulate a hand sanitizer with excellent sterilization power. | [56] |
| Plant antibacterial mite-killing agent, to be prepared accordingly, and it can be used as a daily essential | 2021 | CN112868678A | China | *Thymus vulgaris*, rosemary, *Sophora flavescens*, *Folium artemisiae argyi*, licorice, and dandelion | *E. coli*, *S. aureus*, and *C.'albicans* | The active molecules of the plant extract can act on the brain nerve cells of the mites, thus stimulating the brain neurons, and enabling the mites to enter a deep sleep; this will achieve the effect of efficiently killing the mites. | [57] |
| Composition for improving antibacterial, anti-inflammatory, antiviral, and immune functions, comprising the extract of *ligularia stenocephala* as an active ingredient | 2020 | WO2021182661A1 | South Korea | *Ligularia stenocephala* | *E. coli*, *P. aeruginosa*, *Aspergillus niger*, *Staphylococcus hominis*, *Bacillus subtilis*, and *Streptococcus pneumoniae* | It may be offered as a health-functional food composition or a pharmaceutical composition that enhances antibacterial, anti-inflammatory, and immune functions. | [58] |

**Table 1.** *Cont.*

| Title | Date | Patents | Country | Plants | Bacteria | Application | References |
|---|---|---|---|---|---|---|---|
| Plant extract compositions and methods to make and use plant extract compositions | 2020 | US20210386074A1 | United States | Ginger, green coffee, rosemary, and honeysuckle | *Klebsiella aerogenes*, and *S. aureus* | The extract composition of the present invention may have general or broad-spectrum disinfectant efficacy. | [59] |
| Plant extract hydrolysates and an antibacterial product containing plant extract hydrolysates | 2009 | US9138451B2 | United States | *Equiseti, Juglandis, Millefolii, Quercus, Taraxaci, Althaeae, Matricariae, Centaurium, Levisticum, Rosmarinus, Angelica(e), Artemisia, Astragalus, Leonurus, Salvia, Saposhnikovia, Scutellaria, Siegesbeckia, Armoracia, Capsicum, Cistus, Echinacea, Echinacea, Galphimia,* and *Hedera* | *S. aureus, S. epidermidis, S. pyogenes, S. pneumoniae, Streptococcus mutans,* and *Haemophilus influenzae* | It can be used to produce agents with antibacterial effects against severe infections. | [60] |
| Antibacterial composition comprising plant extracts | 2002 | WO2003035093A1 | South Korea | *Foeniculum vulgare, Illicium verum, Asarum heterotropoides, Cinnamomum* | *Candida* and *Trichophyton* sp. | An antifungal composition that is safe for skin and has superior antifungal activity. It can be applied as a cleaner, a treating agent for dermatomycosis, such as athlete's foot, a disinfectant, among other uses. | [61] |

The antibacterial agents are usually used as a pharmaceutical product to treat specific infections depending on the bioactive compounds. Applications in medicine are presented in Figure 2. To understand the therapeutic effects, several studies identified the synergy or antagonistic effects between different compounds to isolate the single bioactive compound from the complete extract [62]. The antimicrobial molecules in the propolis extract studied by Grecka et al. [63] are compounded by well-known flavonoids (pinocembrin, chrysin, and galangin) which work in harmony against several microorganisms. Results have shown that they exhibit a synergistic effect against Gram-positive and Gram-negative bacteria. Furthermore, Rybczyńska-Tkaczyk et al. [64] combined natural compounds with antimicrobial effects to create cosmetics with polyphenolic compounds, which may also exhibit antioxidant and anti-inflammatory effects. Moreover, some plant extracts exhibited significant antibacterial effects against biofilms. The efficacy of these extracts revealed their potential as drug candidates for eradicating pathogenic bacteria [65]. Häsler et al. [66] discussed extraction technologies that can be used in biomedical applications. The preservation of the biologically active compounds depends on the extraction parameters. For example, water is a biocompatible solvent that is commonly used for cosmetic and medical purposes. Other environmentally friendly options are deep eutectic liquids, such as methanol and ethanol, which can replace organic solvents. Moreover, the conventional maceration method is easy and gentle, but extraction effectiveness can be limited [67]. For this reason, other advanced approaches are suggested, such as ultrasound- and microwave-assisted extraction, as they have more benefits in terms of obtaining a high extraction yield. Finally, the plant extract formulations may depend on the applications and the properties of the extract.

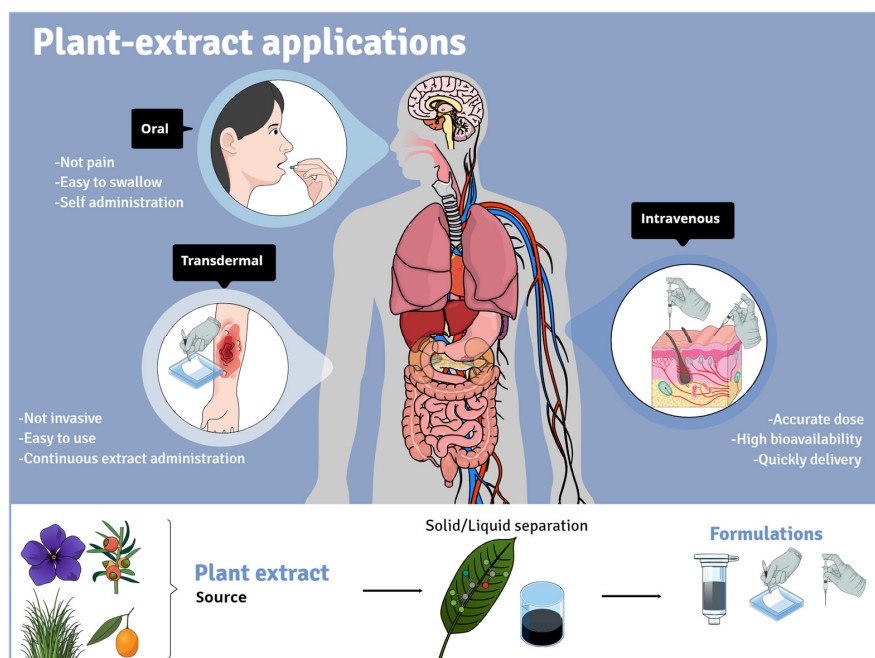

**Figure 2.** Plant extract administration via oral, transdermal, and intravenous methods (https://mindthegraph.com/ accessed on 3 May 2023).

Several bacterial infections have been treated with antibacterial extracts incorporated into gels, creams, microspheres, nanoparticles, and hydrogels. For instance, Raju and Jose [68] evaluated the efficacy of a novel topical gel of neem extract using microspheres as a drug delivery system; the gel produced excellent antibacterial effects. The topical application of antimicrobials offers greater advantages at the site of infection. Iraqui et al. [69] reported on an antimicrobial gel containing *Cassia alata* L. The gel exhibited in vivo wound healing potential due to its significant antibacterial and antifungal activity. Similarly, Popova et al. [70] described a synergistic combination of plant extracts and silver nanoparti-

cles in cream; in the in vivo clinical trials, the cream exhibited significant therapeutic effects on skin diseases, such as antimicrobial action and regenerative effects on tissues. Another structure used for extract-controlled delivery is hydrogels, Nowak et al. [71] presented hydrogels loaded with *Epilobium angustifolium* L. extract; these exhibited antibacterial effects on dermatological diseases. In addition, *Piper crocatum* was successfully encapsulated in polyvinyl alcohol, it showed antibacterial activity in Gram-positive and Gram-negative bacteria, and it can be applied to biomedical devices. Indeed, oral capsules containing *Cola nitida* extract were described by Owusu et al. [72], and they can be used in standard dosages for the management of diarrhea. Finally, the methods that use plant extracts improve biocompatibility and extract-controlled delivery, they decrease toxicity, and enhance biological properties.

There are formulation studies available that used animal models, and some studies have started to use extracts to test for clinically relevant effects in humans. For example, Chaerunisaa et al. [73] demonstrated that *Cassia fistula* extract provided promising antibacterial activities through in vivo tests on female rats, reporting no alterations to the biochemical parameters of the liver and kidneys of the animal models. Moreover, the zebrafish model was used to evaluate the antimicrobial effects of *Lemna minor* plant extracts; it produced excellent results and the safety of the extracts and treatment of bacterial septicemia in vivo were also evaluated [74]. There are approximately 180 clinical trials (www.clinicaltrials.gov, accessed on 3 May 2023) in the world that have tested plant extracts, and different combinations of those extracts, in order to treat a variety of diseases (including respiratory diseases, joint diseases, gastrointestinal diseases, etc.). However, no clinical trials testing plant extracts in humans to combat bacterial diseases have been conducted. This suggests a lack of safety and efficacy data, which has prevented tests on plant extracts to investigate their ability to fight human bacterial diseases.

## 4. Honey and Propolis

Since ancient times, bee products have been recognized for the variety of their biological properties, among which, their antimicrobial activity stands out. In fact, the antibacterial activity of honey was an important finding that was first scientifically described in 1892 by the Dutch scientist Van Ketel [75]. However, other bee-related products (such as propolis) have also been recognized for their biological properties and antimicrobial activities [76,77].

In the case of honey, its antimicrobial properties have been specifically associated with two groups of factors known as (i) peroxide-dependent factors and (ii) non-peroxide-dependent factors. The peroxide-dependent factors of honey are precisely related to the content of hydrogen peroxide ($H_2O_2$) that accumulates in it. $H_2O_2$ is produced in honey due to the action of the glucose oxidase enzyme (produced by the bee) on glucose, which produces gluconic acid and $H_2O_2$ as a by-product of this reaction (Figure 3) [78]; this acts as a sporicidal antiseptic that sterilizes honey and endows it with antibacterial properties against various pathogens.

**Figure 3.** Chemical pathway of the formation of gluconic acid and hydrogen peroxide ($H_2O_2$) in honey.

On the other hand, within the non-peroxide components, osmolality stands out [78]. Honey is a supersaturated solution of sugars, which comprise approximately 80% of its composition. Thus, the osmotically active nature of the sugars causes the dehydration of the bacterial cell, and therefore, its death (Figure 4) [78,79]. In addition to its sugar content, other non-peroxide factors are also important, such as the low pH of honey (between 3.2–4.5) which acts as an inhibitor of various pathogenic bacteria. This acidity is caused by the accumulation of the aforementioned gluconic acid, and other organic acids [78]. There are also non-peroxide factors derived from the floral origin of honey, which include methylglyoxal as the main antimicrobial factor in

Manuka honey, other minor components of honey such as phenolic compounds (i.e., flavonoids and phenolic acids), and some unknown floral components. In fact, it has been proposed that the floral origin of honey plays a fundamental role, not only in terms of its physicochemical properties, but also in terms of its antimicrobial activity [80]. Phenols, flavonoids, terpenes, and alkaloids are also included in the group of antimicrobial-related compounds [81], wherein flavanols are one of the most abundant flavonoids present in food (such as in honey and propolis). Flavanols are well-known for their potent antimicrobial activity against Gram-positive and Gram-negative pathogens, including resistant strains [32]. Moreover, little is known about the types of alkaloid compounds in the floral origins of numerous honey products. However, in 2021, Jaktaji and Ghalamfarsa evaluated the interactions between three monofloral honeys (Avishan, Gavan, and Konar) and ciprofloxacin against *E. coli* [82]. This study demonstrated that all three honey–ciprofloxacin combinations reduced the viability of MG1655 and M1 *E. coli* strains to a greater extent than ciprofloxacin alone. Moreover, the combination of these honeys and the alkaloid extract of *Sophora alopecuroides* enhanced the anti-pump activity and reduced the oxidative stress response of the *E. coli*. Recently, Jaktaji and Koochaki evaluated the in vitro activity of honey and the alkaloid extract of *Sophora alopecuroides* in combination with antibiotics against four biofilm-producing *P. aeruginosa* isolates [83]. This study revealed the synergistic effect of alkaloid extract honey in combination with ciprofloxacin against all *P. aeruginosa* isolates, and it showed a significant reduction in terms of antibiotic resistance and expression of the *mexA* gene [83]. Both studies demonstrated the importance of alkaloids from plant extracts and honeys as sources of antimicrobial agents, and the importance of their combination with standard drugs when working against MDR and biofilm-associated pathogens [82,83].

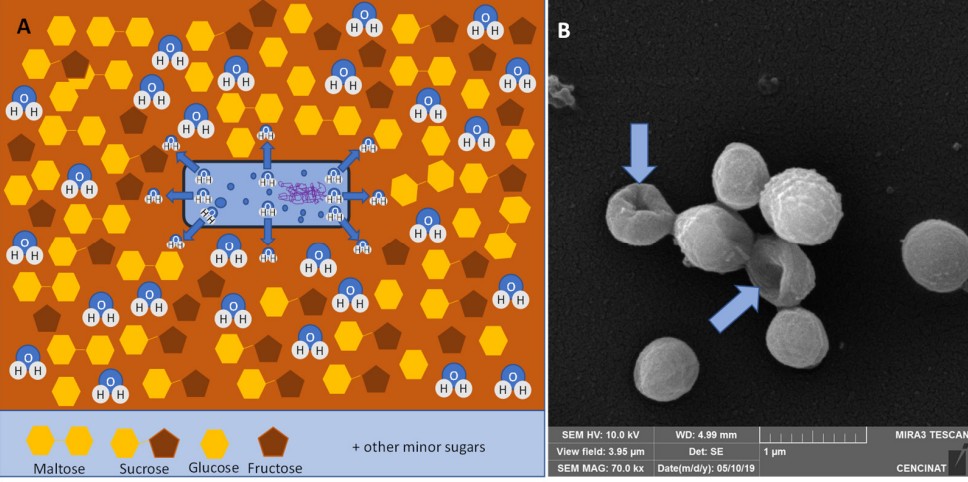

**Figure 4.** Osmotic mechanism related to the antibacterial activity of honey. (**A**) Proposed mechanism of osmotic action mediated by honey sugars, as shown by the blue arrows. (**B**) *S. aureus* cells treated with honey showed cell collapse caused by dehydration, as indicated by the cells with the blue arrows (photos obtained by the authors of the present study).

On the other hand, there are bee-derived factors, such as the bioactive peptide, defensin-1, and other unknown bee components that can pass from the bee to the honey during the honey production process which cause antimicrobial activity [78].

The spectrum of antibacterial activity for honey is broad, both from the point of view of the flora and geographical origin, and the pathogens that have shown susceptibility to this antibacterial activity. Thus, honey of various floral origins, both polyfloral and monofloral, and honey from different geographical origins, have proven to be effective, not only for inhibiting the growth of pathogenic bacteria in the planktonic state, but also for inhibiting bacterial biofilm formation and eradicating preformed biofilm [84]. Among the most studied pathogens are the Gram-positive *S. aureus* and MRSA, and the Gram-negative *P. aeruginosa*, *K. pneumoniae* (including *K. pneumoniae carbapenemase*, KPC), and *E. coli* (Table 2).

**Table 2.** Geographical and floral origin of honey, as well as the main pathogens and methods used in the study of the antimicrobial activity of honey.

| Geographical Origin | Floral Origin | Bacterial Strain | Analytical Method | References |
|---|---|---|---|---|
| Australia | *E. marginata, E. patens, E. platypus, E. wandoo Banksia* spp., *Callistemon* spp., *Corymbia calophylla, Leptospermum subtenue*, and *Leptospermum scoparium* | *S. aureus* ATCC 25923, ATCC 29213 and NCTC 10442, *S. epidermidis* ATCC 11047, *E. faecalis* ATCC 29212 and ATCC 51299, *A. baumannii* ATCC 7844, *E. coli* ATTC 25922, *P. aeruginosa* ATCC, 27853, and *Salmonella enterica* subsp. *enterica* serovar Typhimurium ATCC 13311 | Kirby–Bauer Test Minimal inhibitory concentration (MIC) and minimal bactericidal concentration (MBC) | [85] |
|  | *E. macrorrhyncha* | *Alcaligenes faecalis, Citrobacter freundii, E. coli, K. aerogenes, K. pneumoniae, Mycobacterium phlei, Salmonella enterica* subsp. *enterica* serovar California, *Salmonella enterica* subsp. *enterica* serovar Enteritidis, *S.* Typhimurium, *Shigella sonnei, S. aureus* and *S. epidermidis, Serratia marcescens*, and *C. albicans* | MIC (agar dilution method) | [86] |
| Colombia | Polyfloral honey | *E. coli* ATCC 25922, *S.* Typhimurium ATCC 14028, and *Listeria monocytogenes* ATCC 19118 | Agar diffusion technique | [87] |
| Cuba | *Turbina corymbosa* (L.) Raf, *Ipomoea triloba* L., *Avicennia germinans* Jacq., *Govania polygama* (Jack) Urb, and *Lysiloma latisiquum* (L.) Benth | *B. subtilis* ATCC 6633 and *S. aureus* ATCC 25923 | Minimum active dilution (MAD) via the agar incorporation technique | [88] |
|  | Polyfloral honey | Clinical isolates: *S. aureus* 13, *S. epidermidis* 35, *S. pneumoniae* 9, *S. pyogenes* 12, *S. pyogenes* C-105, *S. pyogenes* m46, *Streptococcus agalactiae* 1357, *Streptococcus mitis* 22, *Streptococcus oralis* 1235, *Streptococcus anginosus* 2513, *Streptococcus parasanguinis* 2761, *Streptococcus salivarius* 14, *Streptococcus gordonii* 143, *E. faecalis* 212, *E. faecium* 17, *L. monocytogenes* 49, *Enterobacter cloacae* 19902, *C. freundii* 55, *Salmonella enterica* subsp. *enterica* serovar Fyris 3813, *S. marcescens* 28315, *A. baumanii* 8, *K. pneumoniae* 15, *P. aeruginosa* 24. *E. coli* 23, *Proteus mirabilis* 112, and *C. albicans* 18 | MAD via the agar incorporation technique | [89] |
|  | Polyfloral honey | Clinical isolates: *S. aureus* 13, *S. epidermidis* 35, *S. pneumoniae* 9, *S. pyogenes* 12, *S. pyogenes* C-105, *S. pyogenes* m46, *S. agalactiae* 1357, *S. mitis* 22, *S. oralis* 1235, *S. anginosus* 2513, *S. parasanguinis* 2761, *S. salivarius* 14, *S. gordonii* 143, *E. faecalis* 212, *E. faecium* 17, *L. monocytogenes* 49, *E. cloacae* 19902, *C. freundii* 55, *S.* Fyris 3813, *S. marcescens* 28315, *A. baumanii* 8, *K. pneumoniae* 15, *P. aeruginosa* 24. *E. coli* 23, *P. mirabilis* 112, and *C. albicans* 18 | MAD via the agar incorporation technique Inhibition of biofilm formation and removal of preformed biofilm assay Transmission Electron Microscopy (TEM) for morphology analysis | [90] |

**Table 2.** *Cont.*

| Geographical Origin | Floral Origin | Bacterial Strain | Analytical Method | References |
|---|---|---|---|---|
| Ecuador | *Eucalyptus* spp. | *S. aureus* ATCC 25923, *S. pyogenes* ATCC 19615, *P. aeruginosa* ATCC 27853, *E. coli* ATCC 25922, and *C. albicans* ATTC 90028 | MAD via the agar incorporation technique | [91] |
| | | *S. aureus* CAMP and *K. pneumoniae* KPC 609803 | Inhibition of biofilm formation and removal of preformed biofilm assays | [80] |
| | *Eucalyptus* spp. | *S. aureus* MRSA ATCC 2592, *P. aeruginosa* ATCC 2785, and *S. aureus* MRSA S21 (clinical isolate) | Inhibition of biofilm formation and removal of preformed biofilm assay TEM for morphology analysis | [92] |
| | *Eucalyptus* spp. | *S. aureus*, MRSA S21 (clinical isolate), and *P. aeruginosa* P28 clinical isolate) | Inhibition of biofilm formation and removal of preformed biofilm assay TEM for morphology analysis | [79] |
| | *Persea americana* | *S. aureus* CAMP and *K. pneumoniae* KPC 609803 | Inhibition of biofilm formation and removal of preformed biofilm assay | [80] |
| | *Brassica napus* | *S. aureus* CAMP and *K. pneumoniae* KPC 609803 | Inhibition of biofilm formation and removal of preformed biofilm assay | [80] |
| Iran | *Eucalyptus* spp. | *E. coli* ATCC 25922, *P. aeruginosa* ATCC 27853, *S. aureus* ATCC 25923, and *E. faecalis* ATCC 11700 | Kirby–Bauer Test | [93] |
| Italy | *Eucalyptus* spp. | *S. aureus* subsp. *aureus* ATCC 9144, and *P. aeruginosa* ATCC 27853 | Kirby–Bauer Test | [94] |
| Kenya | Polyfloral honey | Clinical isolates: *S. aureus* 13, *S. epidermidis* 35, *S. pneumoniae* 9, *S. pyogenes* 12, *S. pyogenes* C-105, *S. pyogenes* m46, *S. agalactiae* 1357, *S. mitis* 22, *S. oralis* 1235, *S. anginosus* 2513, *S. parasanguinis* 2761, *S. salivarius* 14, *S. gordonii* 143, *E. faecalis* 212, *E. faecium* 17, *L. monocytogenes* 49, *E. cloacae* 19902, *C. freundii* 55, *S.* Fyris 3813, *S. marcescens* 28315, *A. baumanii* 8, *K. pneumoniae* 15, *P. aeruginosa* 24, *E. coli* 23, *P. mirabilis* 112, and *C. albicans* 18 | MAD via the agar incorporation technique Inhibition of biofilm formation and removal of preformed biofilm assay TEM for morphology analysis | [90] |
| Mauritius | *Eucalyptus* spp. | *E. coli* (clinical isolate), *E. coli* ATCC 25922, *Proteus* spp. (clinical isolate), *P. mirabilis* ATCC 12453, *Pseudomonas* spp. (clinical isolate), *P. aeruginosa* ATCC 27853, *Klebsiella* spp. (clinical isolate), *Streptococcus* spp. (clinical isolate), and *S. epidermidis* ATCC 35984 and ATCC 14990 | Kirby–Bauer Test | [95] |

**Table 2.** *Cont.*

| Geographical Origin | Floral Origin | Bacterial Strain | Analytical Method | References |
|---|---|---|---|---|
| New Zealand | *Leptospermum scoparium* | Clinical isolates: *S. aureus* 13, *S. epidermidis* 35, *S. pneumoniae* 9, *S. pyogenes* 12, *S. pyogenes* C-105, *S. pyogenes* m46, *S. agalactiae* 1357, *S. mitis* 22, *S. oralis* 1235, *S. anginosus* 2513, *S. parasanguinis* 2761, *S. salivarius* 14, *S. gordonii* 143, *E. faecalis* 212, *E. faecium* 17, *L. monocytogenes* 49, *E. cloacae* 19902, *C. freundii* 55, *S.* Fyris 3813, *S. marcescens* 28315, *A. baumanii* 8, *K. pneumoniae* 15, *P. aeruginosa* 24. *E. coli* 23, *P. mirabilis* 112, and *C. albicans* 18 | MAD via the agar incorporation technique | [89] |
| | *Leptospermum scoparium* var. *Incanum*, *Leptospermum scoparium* var. *incanum* + *Kunzea ericoides* *Leptospermum scoparium* var. *incanum* + *Kunzea ericoides*, and *Trifolium* spp. | *S. aureus* NCTC 8325 and ATCC 25923, *S. aureus* HA-MRSA, and *S. aureus* CA-MRSA | Inhibition of biofilm formation and removal of preformed biofilm assays | [96] |
| Pakistan | *Ziziphus mauritiana, Azadirachta indica, Ziziphus spina-christi, Citrus sinensis*, and *Brassica nigra* | *Salmonella enterica* subsp. *enterica* serovar Typhi | MBC, MIC, and agar well diffusion assays | [97] |
| Poland | *Prunus spinosa* L., Polyfloral honey, *Salix* spp., *Brassica napus* L., *Phacelia tanacetifolia* Benth., *Solidago vigaurea* L., and *Helianthus* spp. | *E. coli* D31 (CGSC 5165), *Bacillus circulans* ATCC 61; *S. aureus*, 1-KI (clinical isolate), *P. aeruginosa* (ATCC 27853), *P. aeruginosa* 02/18 (clinical isolate), *A. niger* 71, *Saccharomyces cerevisiae*, and *C. albicans* | MAD via the agar incorporation technique | [98] |
| Slovakia | *Crataegus laevigata, Abies alba* Mill, and *Robinia pseudoacacia* | *P. mirabilis* and *E. cloacae* | MAD via the agar incorporation technique Removal of preformed biofilm assay | [99] |
| | *Robinia pseudoacacia, Rubus* spp., *Brassica napus, Rubus idaeus*, and *Phacelia* spp. | *P. aeruginosa* CCM1960 and *S. aureus* CCM4223 | MIC and MBC assays | [100] |
| Spain | *Eucalyptus* spp. | *S. aureus* and MRSA (clinical isolate), *S. pyogenes, E. coli*, and *P. aeruginosa* (clinical isolate) | Disk–plate diffusion method | [101] |
| Turkey | *Nigella sativa* L. | *E. coli* ATCC 25,922, *E. faecalis* ATCC 29,121, *S. aureus* ATCC 6538, *S. enteric* subsp. *enterica* ATCC 14,028/363–154, *B. subtilis* B209, *Bacillus cereus*, and *L. monocytogenes* ATCC 7677 | Disc diffusion and MIC assays | [102] |

Propolis is another bee product that has shown important antimicrobial properties [103]. Propolis, also known as bee glue, is a sticky resinous substance that bees collect from living plants during their nectar and pollen-collecting activities. Its composition is complex, formed mainly by vegetable resins (50%), waxes (30%), aromatic and essential oils (10%), pollen (5%), and other organic compounds (5%). This composition is highly influenced by its floral origin, as is its color, which can range from green to reddish to brown [76]. Regarding its antimicrobial activity, it must be considered through two mechanisms of action. (i) The first is related to the direct action on the microorganism. This action is mainly related to the action of the propolis components on the permeability of the cell membrane of microorganisms, the disruption to the membrane potential, and the production of adenosine triphosphate (ATP), as well as the reduction in bacterial mobility [104]. In fact, the antimicrobial activity of propolis has been reported to be more effective on Gram-positive bacteria than Gram-negative bacteria [103]. This has been explained by the typical structure of the outer membrane of Gram-negative bacteria and the production of hydrolytic enzymes that break down the active components of propolis [105]. The second (ii) mechanism is related to its ability to stimulate the immune system; this results in the activation of the body's natural defenses [104]. Although the antibacterial activity of propolis has been tested in a varied number of microorganisms, there is a group of bacterial strains that has been more widely analyzed in terms of the strains' susceptibility to propolis extracts. The ten most tested bacteria for their susceptibility to propolis extracts from different geographic origins include *E. coli*, *S. aureus*, *Salmonella* spp., *P. aeruginosa*, *Yersinia enterocolitica*, *Enterococcus* spp., *P. mirabilis*, *K. pneumoniae*, *S. mutans*, and *S. epidermidis* [103]. However, studies examining the antibacterial activity of propolis do not only focus upon the aforementioned bacterial groups. Analyses concerning the antibacterial activity of propolis cover a wide group of propolis extracts from different regions of the world and a wide range of microorganisms (Table 3).

Polyphenols and terpenoids are the main components of propolis that have been identified as being responsible for propolis' antimicrobial activity [106]. Their profiles are closely related to resins and balms of floral origin, as well as the climatic conditions and geographical area where the plants used to produce it grow. Thus, in Europe, North America, and Asia (temperate zone), polyphenolic profiles are characterized by high levels of flavonoids (mainly flavones and flavanones) and low levels of phenolic acids, whereas in tropical zones, propolis shows a more complex composition, with prenylated flavonoids, prenylated *p*-coumaric acids, and lignans, among others [76]. Several of these compounds have been identified as components of propolis and they exhibit a high degree of antimicrobial activity. An example of such a compound is artepillin C (3,5-diprenyl-p-coumarid acid), a prenyl derivative of *p*-coumaric acid that can be isolated from propolis. Extracts rich in artepillin C showed a high degree of antibacterial activity against MRSA, as well as against anaerobic bacteria such as *Porphyromonas gingivalis*, where it exhibited an effective bacteriostatic effect [103] Similarly, other prenyl derivatives reported in propolis, such as 3-prenyl-cinnamic acid allyl ester and 2-dimethyl-8-prenylchromene, have also shown similar antimicrobial activities [107]. Moreover, not only are prenyl derivatives responsible for the antimicrobial activity of propolis, but another abundant group in this product has also been found to have similar properties; this is the flavonoid group. Flavonoids represent a group of important polyphenolic components present in propolis, which are closely related to the high functionality of this bee product [76]. This flavonoid group includes chrysin, pinocembrin, apigenin, galangin, kaempferol, kaempferide, quercetin, tectochrysin, pinostrobin, and others [76]. Pinocembrin isolates were shown to be highly effective against *Streptococcus sobrinus*, *S. mutans*, *S. aureus*, *E. faecalis*, *L. monocytogenes*, *P. aeruginosa*, and *K. pneumoniae*. Furthermore, isolated apigenin was effective against *P. aeruginosa*, *K. pneumoniae*, *S.* Typhimurium, *P. mirabilis*, and *K. aerogenes*. Similarly, the synergistic antibacterial effect of apigenin, together with beta-lactam antibiotics, was also observed against MRSA [45]. In addition, apigenin and ceftazidime also exhibited a synergistic antibacterial effect against ceftazidime-resistant *E. cloacae* [103]. Propolis, as a material

composed largely of plant secretions, is a rich source of phenyl acids, such as cinnamic acid and esters. Several studies have reported the antimicrobial activity of cinnamic acid against various microorganisms, such as *Aeromonas* spp., *Vibrio* spp., *E. coli*, *L. monocytogenes*, *Mycobacterium tuberculosis*, *Bacillus* spp., *Staphylococcus* spp. *S. pyogenes*, *Micrococcus flavus*, *P. aeruginosa*, *S.* Typhimurium, *E. cloacae*, and *Yersinia ruckeri* [103]. However, not only do these constituents separately contribute to the antimicrobial activity of propolis, but their interactions may be another mechanism by which antimicrobial activity may be enhanced. Thus, the ethanolic extract of propolis, which contains high concentrations of kaempferide, artepillin-C, drupanin, and the phenolic acid, *p*-coumaric acid, showed significant antibacterial activity against *S. aureus*, *Staphylococcus saprophyticus*, *L. monocytogenes*, and *E. faecalis* [108].

**Table 3.** Geographical origin and the main pathogens used in the study of the antimicrobial activity of propolis using the MIC assay.

| Geographical Origin | Bacterial Strain (Gram-Positive) | References | Bacterial Strain (Gram-Negative) | References |
|---|---|---|---|---|
| Australia | *S. aureus* ATCC 25923 | [109] | *K. pneumoniae* ATCC 13883 | [109] |
| Brazil | *B. subtilis* ATCC 6633, *Enterococcus* spp., *E. faecalis* ATTC 29212, ATCC 43300 and ESA 553, *Micrococcus luteus* ATCC 10240, *S. aureus* ATCC 6538, ATCC 43300, ATCC 25923, SA 10 and ESA 654, *S. epidermidis* ATCC 12228 and ESA 675, *S. mutans*, and *S. pyogenes* | [110–115] | *E. coli* ATCC 8739, ATCC 25922 and EC06, *K. pneumoniae* ATCC 4352 and ESA 154, *P. mirabilis* ATCC 43300 and ESA 37, *P. aeruginosa* ATCC 25853, ATCC 15442, PA 24 and ESA 22, and *Salmonella* spp. | [111,113–115] |
| Bulgaria | *S. aureus* ATCC 209 | [116] | *E. coli* WF | [116] |
| Chile | *S. aureus* ATCC 25923 and *S. pyogenes* ISP 364-00 | [117] | *E. coli* ATCC 25922 and *P. aeruginosa* ATCC 27853 | [117] |
| Czech Republic | *S. aureus* ATCC 29213, ATCC 25923 and ATCC 977, *S. epidermidis* ATCC 14990, *S. aureus* MRSA/NCTC, *S. saprophyticus* ATCC 15305, *S. oralis* ATCC 35037, *B. subtilis* ATCC 6051, *Enterococcus* spp., *S. agalactiae* ATTC 27956, *S. pneumoniae* ATCC 49619, and *S. pyogenes* ATCC 12344 | [118] | *A. baumani, Burkholderia cepacia, E. cloacae* ATCC 700323, *E. coli* O157:H7, *H. influenzae* ATCC 49747, *K. pneumoniae* ATCC 700603., *P. aeruginosa* ATCC 27853, *Salmonella* spp., *Shigella flexneri*, and *Y. enterocolitica* ATCC 9610 | [118] |
| Germany | *S. aureus* ATCC 29213, ATCC 25923 and ATCC 977, *S. epidermidis* ATCC 14990, *S. aureus* MRSA/NCTC, *S. saprophyticus* ATCC 15305, *S. oralis* ATCC 35037, *B. subtilis* ATCC 6051, *Enterococcus* spp., *S. agalactiae* ATTC 27956, *S. pneumoniae* ATCC 49619, and *S. pyogenes* ATCC 12344 | [118] | *A. baumani, B. cepacia, E. cloacae* ATCC 700323, *E. coli* O157:H7, *H. influenzae* ATCC 49747, *K. pneumoniae* ATCC 700603., *P. aeruginosa* ATCC 27853, *Salmonella* spp., *S. flexneri*, and *Y. enterocolitica* ATCC 9610 | [118] |
| Greece | *S. aureus* ATCC 25923 and *S. epidermidis* ATCC 12228 | [119] | *E. cloacae* ATCC 13047, *E. Coli* ATCC 25922, *P. aeruginosa* ATCC 227853, and *K. pneumoniae* ATCC 13883 | [119] |
| India | *S. aureus* ATCC 6538P | [120] | – | – |
| Ireland | *S. aureus* ATCC 29213, ATCC 25923 and ATCC 977, *S. epidermidis* ATCC 14990, *S. aureus* MRSA/NCTC, *S. saprophyticus* ATCC 15305, *S. oralis* ATCC 35037, *B. subtilis* ATCC 6051, *Enterococcus* spp., *S. agalactiae* ATTC 27956, *S. oralis, S. pneumoniae* ATCC 49619, and *S. pyogenes* ATCC 12344 | [118] | *A. baumani, B. cepacia, E. cloacae* ATCC 700323, *E. coli* O157:H7, *H. influenzae* ATCC 49747, *K. pneumoniae* ATCC 700603, *P. aerugino-sa* ATCC 27853, *Salmonella* spp., *S. flexneri*, and *Y. enterocolitica* ATCC 9610 | [118] |

**Table 3.** *Cont.*

| Geographical Origin | Bacterial Strain (Gram-Positive) | References | Bacterial Strain (Gram-Negative) | References |
|---|---|---|---|---|
| Italy | – | – | *Campylobacter jejuni* (clinical isolate) and *P. aeruginosa* P1242 | [121,122] |
| Korea | *S. mutans* ATCC 25175, *S. sobrinus* ATCC33478, *S. mutans* KCOM 1088, KCOM 1091, KCOM 1092, KCOM 1095, KCOM 1097, KCOM 1111, KCOM 1112, KCOM 1113, KCOM 1116, KCOM 1117, KCOM 1118, KCOM 1123, KCOM 1124, KCOM 1126, KCOM 1127, KCOM 1128, KCOM 2762, KCOM 1136, KCOM 1137, KCOM 1139, KCOM 1142, KCOM 1143, KCOM 1145, KCOM 1146, KCOM 1197, KCOM 1200, KCOM 1201, KCOM 1202, KCOM 1203, KCOM 1207, KCOM 1208, KCOM 1209, KCOM 1212, KCOM 1214, KCOM 1217, KCOM 1219, KCOM 1226 (clinical isolates), and *S. sobrinus* KCOM 1061, KCOM 1150, KCOM 1151, KCOM 1152, KCOM 1153, KCOM 1157, KCOM 1158, KCOM 1159, KCOM 1185, KCOM 1191, KCOM 1193, KCOM 1196, KCOM 1221, KCOM 1228, and KCOM 1218 (clinical isolates) | [123] | – | – |
| Morocco | *S. aureus* ATCC 6538 and MRSA 2, 15, and 16 (clinical isolates) | [124] | – | – |
| Oman | *S. aureus* ATCC 209 | [116] | *E. coli* WF | [116] |
| Poland | *S. aureus* ATCC 25923 and *S. aureus* (clinical isolates) | [125] | – | – |
| Slovakia | *B. cereus* WSBC 10530, *S. aureus* ATCC 25923, *S. aureus* Z MJ346, *S. pyogenes* Z M494, *E. faecalis* Z MJ90, *L. monocytogenes* Z M58, and *L. monocytogenes* Z M70 | [126] | *E. coli* ATCC 11229, *E. coli* O157:H7 Z MJ128, *S. typhimurium* ATCC 14028, *S. enteritidis* Z M138, *C. coli* ATCC 33559, *C. coli* 2235, *C. coli* 3341-05, *C. jejuni* ATCC 33560, *C. jejuni* NCTC 11168, *C. jejuni* 375-06, and *C. jejuni* 3552 | [126] |
| Turkey | *S. mutans* ATCC 25175, *S. aureus* 6538-P, *S.sobrinus* ATCC 33478, *S. epidermidis* ATCC 12228, *E. faecalis* ATCC 29212, and *M. luteus* ATCC 9341 | [127] | *P. aeruginosa* ATCC 27853, *E. coli* ATCC 11230, *S.* Typhimurium CCM 5445, and *K. aerogenes* ATCC 13048 | [127] |

## 5. Bacteria as a Source of Alternative or Complementary Treatments (Prebiotics, Probiotics, Symbiotics, and Postbiotics)

Since ancient times, bacteria and their products have been used to benefit humans, both in terms of health and food [128]. Prebiotics and probiotics constitute the two main research fields concerning bacterial applications for human benefit. Indeed, the term 'probiotic' is actually derived from Greek/Latin and means 'for life'. Prebiotics are characterized by a group of compounds that are metabolized by microbiota which enhance the growth of probiotic bacteria [128,129]. Over the years, a great deal of research has been conducted on probiotics, and therefore, many definitions have been suggested, however, in 2014, the Food and Agriculture Organization (FAO) and the World Health Organization (WHO) of the United Nations defined probiotics as "live microorganisms which when administered in adequate amounts confer a health benefit on the host" [130]. Likewise, the term 'prebiotic' has evolved over the years; however, in 2017, it was defined as "a substrate that is selectively utilized by host microorganisms conferring a health benefit" [131]. A prebiotic substrate must neither be hydrolyzed nor absorbed by the host's mucosal barrier (such as the gastrointestinal tract), and it must be selectively metabolized by one (or a limited number of) potentially beneficial bacteria that reside in the microbiota. Currently, probiotic and prebiotic concepts have expanded due to recent developments and findings in microbiome research. High-throughput sequencing studies [132,133] have allowed us to improve our knowledge of the composition of microbiota and identify additional substances influencing microbial colonization [131].

Generally, the most studied and used probiotic bacteria belong to the genera *Lactobacillus* and *Bifidobacterium*, such as *Lactobacillus acidophilus*, *L. rhamnosus*, *L. johnsonii*, *L. casei*, *L. delbrueckii* sp. *bulgaricus*, *L. reuteri*, *L. brevis*, *L. fermentum*, *L. plantarum*, *Bifidobacterium bifidum*, *B. adolescentis*, *B. animalis*, *B. infantis*, and *B. thermophilum* [134–139]. However, other bacteria and yeasts have also been recognized for their probiotic properties, such as *B. subtilis*, *Propionibacterium* spp., *and S. cerevisiae* var. Boulardii [140–143]. The mechanism of action of probiotics relies on their metabolites, the colonization of the host's barriers (i.e., skin and mucosal epithelia), competition for nutrients, and production of antimicrobial agents [144], such as lactic acid, hydrogen peroxide, bacteriocins, bacteriocin-like proteins, and biosurfactants [145–147]. Probiotic bacteria are known to produce small molecular metabolic byproducts that influence the host's biological functions. Although some studies are trying to explain how probiotics protect the host [134,138,148,149], these modulation and support mechanisms have not been fully explored. Moreover, the majority of prebiotics are usually characterized by short-chain carbohydrates which are polymerized ($\geq 2$) and not susceptible to digestion via the host's intrinsic enzymes (pancreatic, intestinal, and other mucosal enzymes) [150–152]. Most compounds classified as prebiotics are fructans, lactulose, xylooligosaccharides (XOS), and mannan oligosaccharides (MOS). Fructans are mainly composed of several units of fructose linked to terminal sucrose, such as inulin, fructooligosaccharides (FOS), and galactooligosaccharides (GOS) [153–155]. Usually classified as health supplements or food products, prebiotics promote the growth of certain probiotic or beneficial bacteria (e.g., lactobacilli, propionibacteria, and bifidobacteria) for the host's well-being, which improve both epithelial and mucosal protection during external stress or contact with primary and opportunistic pathogens. Prebiotics are also able to promote immune system responses, as shown in several studies which have reported beneficial effects in the gut-associated lymphoid tissue (GALT) [154,156,157]. However, over the last decade, findings have also reported on microbiota being found in other places in the host, such as on the skin and pulmonary mucosa, among others [158–161].

Recent and ongoing developments in microbiome science are creating new frontiers for research on probiotics and prebiotics (Table 4) [162,163]; indeed, the scientific community has shown an increased interest in synbiotics and postbiotics [155,164,165]. Synbiotics are defined by a combination of probiotics and prebiotics; for instance, a suitable prebiotic could enhance the probiotic's chance of survival and biological activity. The combination of prebiotics and probiotics could also possess synergistic effects that promote the growth

of existing beneficial bacteria in the host, as well as improving the survival, implantation, and growth of newly added probiotic species/strains [128,166]. Synbiotics have not been extensively studied [167,168] and fewer human clinical trials have been carried out on the effectiveness of synbiotics [169,170]. In fact, further studies on different combinations working against different MDR pathogens are needed to create accurate formulations and to undertake further evaluations. It is important to mention that individual probiotic species and strains have different prebiotic requirements, and furthermore, different probiotic species and strains are more efficient against certain pathogen-related infections [135,138]. Moreover, the term 'postbiotics' is defined as the "preparation of inanimate microorganisms and/or their components that confers a health benefit on the host"; this definition was devised by a scientific panel from the International Scientific Association for Probiotics and Prebiotics (ISAPP) [171]. Recognizing that the term 'postbiotic' means "after life" (not "from life"), postbiotics are characterized by the presence of inanimate microbes and/or physiologically active microbial cellular components (such as cell surface fragments, enzymes, and metabolites) that can contribute to the complexity and functionality of the host's beneficial effects against illness or infectious diseases [171]. These metabolites are recognized as metabolic byproducts that are secreted by the microorganism; they include enzymes, peptides, teichoic or organic acids, polysaccharides, and other compounds. Therefore, postbiotics are derived from probiotic or even non-probiotic cells, thus providing health benefits for the host when applied in adequate quantities and in requisite combinations [172]. However, the health-improving properties or aspects of postbiotics, and their bioactivities, are still unknown or unclear [173]. Currently, it is possible to better understand the advantages of these postbiotic compounds on one's overall health due to the new -omic analyses (such as genomics, transcriptomics, metabolomics, lipidomics, and proteomics) that are currently being examined in several investigations [174,175].

Table 4. Summary of recent in vivo and in vitro studies reporting beneficial effects for the host and the antimicrobial activities of prebiotics, probiotics, synbiotics, and postbiotics.

| Type of Biotics | Compounds and/or Species | Antimicrobial Activity | References |
|---|---|---|---|
| Prebiotics | *Punica granatum* peel extract | In vitro and in vivo *Caenorhabditis elegans* nematode model assays demonstrated a reduction in hemolytic activity and biofilm formation caused by *P. aeruginosa* and they promoted the growth of *B. bifidum* and *L. plantarum* probiotic strains. | [176] |
| | Chitin-glucan (CG) | In vitro assays showed high levels of growth in all bifidobacterial species, particularly the *Bifidobacterium breve* 2L isolate in the in vivo Groningen rat model, which became more abundant in the gut of *B. breve* 2L. | [177] |
| | Phthalyl pullulan nanoparticles (PPNs) | A gut dysbiosis-induced murine model was used, and their restorative effect in the eubiosis microbiota was assessed using the pathogen, *E. coli* K99. | [167] |
| | Ursodeoxycholic acid (UDCA) | In vitro assays demonstrated a reduction in *E. coli* serotype O101:H9 growth, proinflammatory effects in Caco-2 cells, and cell integrity damage. Moreover, in vivo assays used on neonatal mice model also exhibited attenuated colitis symptoms and recovered colonic short-chain fatty acid (SCFA) production. | [178] |
| Probiotics | Two *Lactococcus lactis* subsp. *lactis* strains | In vivo bacterial feeding of these probiotic strains for 30 days in a Swiss albino mice model was conducted, and they improved gut colonization and IgA levels. | [168] |
| | Honey probiotic *B. circulans* isolate | In vitro co-culture of *B. circulans* and *Cutibacterium acnes* significantly suppressed pathogen growth. Moreover, in vivo assays using the ICR mice model, *B. circulans*, generated electrons that inhibited *C. acnes* growth and diminished inflammation. | [179] |
| | *Bacillus amyloliquefaciens* (BA PMC-80) | In vitro co-culture assays of BA PMC-80 and *Clostridioides difficile* demonstrated significant pathogen inhibition; an in vivo hamster model exhibited no toxicity, a less severe infection, and late death. | [180] |
| | Two new strains of *B. subtilis* (CH311 and S3B) | In vitro gut model demonstrated the ability of *B. subtilis* CH311 and S3B to reduce ESBL-*E. coli* titers using 4 log CFU/mL; however, the in vivo murine model showed no reduction in the ESBL-*E. coli* fecal titers. | [181] |
| Synbiotics | *Pediococcus acidilactici* plus phthalyl inulin nanoparticles (PINs) | In vitro antimicrobial activity was tested using a cocultivation assay. A statistical reduction of more than 3 log CFU/mL of *Salmonella enterica* subsp. *enterica* serovar Gallinarum, together with *P. acidilactici* plus PINs, was observed when compared with the control and the PINs or probiotic groups alone. | [182] |
| | Encapsulated *S. cerevisiae* plus *Moringa oleifera* leaf extract (MOLE) | In vivo rabbit model revealed no effects on interleukin-l or IgG and IgA levels, and it showed a significantly higher number of beneficial microbes. Moreover, a significant increase in in vitro inhibitory activities was observed against *E. coli* BA 12296B, *S. aureus* NCTC 10788, *C. albicans* ATCC MYA-2876, *L. monocytogenes* ATCC 19116, and *Salmonella enterica* subsp. *enterica* serovar Senftenberg ATCC 8400. | [183] |
| | PPNs plus *L. plantarum* | A gut dysbiosis-induced murine model was used and the *E. coli* K99 infection was markedly suppressed after several well-known beneficial bacteria, including *Lactobacillus* and *Bifidobacterium*, were incrementally introduced. | [167] |
| | Two *L. lactis* subsp. *lactis* strains with inulin | The Swiss albino mice model exhibited a significant reduction in IgA levels that is comparable with commercial probiotics and prebiotic consortiums on the market. | [168] |

**Table 4.** *Cont.*

| Type of Biotics | Compounds and/or Species | Antimicrobial Activity | References |
|---|---|---|---|
| Postbiotics | Lyophilized cell-free supernatants (LCFS) of *Lactobacillus* isolates | Demonstrated strong inhibition and eradication antibiofilm activities for *A. baumannii* and *E. coli*, and a reduction in nitric oxide production in the RAW 264.7 cell line was observed. | [164] |
| | *L. plantarum* KM1, *L. plantarum* KM2, *Bacillus velezensis* KMU01 postbiotics mixtures 1:1:1 (vol/vol) | NK cell activation was significantly higher in the C57BL/6N mice model, and TNF-α levels in the RAW264.7 cell line was significantly reduced when compared with the LPS. | [184] |
| | Individual LCFS of *L. fermentum*, *L. reuteri*, and *B. subtilis* sp. *natto* in a postbiotic cold cream | All postbiotic cold creams exhibited different degrees of immunomodulatory, anti-inflammatory, and antimicrobial activities in the Sprague Dawley rat model when compared with controls (no treatment and only cold cream). | [185] |
| | Indole-3-carboxaldehyde (3-IAld) (a microbial tryptophan metabolite) | The use of 3-IAld inhalable dry powder demonstrated optimal pulmonary administration and toxicological safety, also reducing aspergillosis scores by acting on the infection and inflammation sites. | [186] |

Legend—CFU: Colony-forming units; IgA: Immunoglobulin A; ICR: Crl:CD1; Vol/vol: Volume/volume; NK: Natural killer; TNF-α: Tumor necrosis factor alpha; LPS: Lipopolysaccharides.

Human microbiota are considered "functional organisms" as a complex community of microorganisms coexist on the host's skin and in mucosal tissues in a healthy balance [134,137]. Microbiota are crucial for the metabolism and immune system regulation, as well as for the prevention of potential pathogen colonization. Thus, an imbalance in human microbiota can be detrimental to the host's equilibrium and can cause a state of dysbiosis [150]. Hence, studies into the rise of emerging MDR pathogens have demonstrated how the use of prebiotics and probiotics can help fight these virulent and antibiotic-resistant pathogens in the near post-antibiotic era. In 2019, Joshi and colleagues studied *Punica granatum* peel extract for its quorum-modulatory potential against two different human-pathogenic bacteria, viz. *Chromobacterium violaceum* and *P. aeruginosa*; it exhibited notable prebiotic potential by promoting the growth of *B. bifidum* and *L. plantarum* probiotic strains [176]. Moreover, the virulent traits of *P. aeruginosa*, such as hemolytic activity and biofilm formation, were negatively affected by this extract in in vitro assays, and its therapeutic efficiency was confirmed as the nematode, *C. elegans*, was also more susceptible to lysis by human sera [176]. Alessandri and colleagues analyzed the effect of chitin-glucan (CG), as a biopolymer of *A. niger*, on one hundred bifidobacterial strains from infant feces and the gastrointestinal tract of adults [177]; the study demonstrated that almost all bifidobacterial species displayed high growth levels in the in vitro assays, in particular, the *B. breve* 2L isolate. When evaluating the colonization of *B. breve* 2L in the mammalian gut via CG stimulation, the in vivo Groningen rat model (*Rattus norvegicus*) exhibited a significant increase in the gut of *B. breve* 2L, thus enhancing the gut colonization/persistence of this strain and suggesting that CG exerts a species specific modulation of the bifidobacterial population that is harnessed by the rat gut [177]. In 2022, He and colleagues studied ursodeoxycholic acid (UDCA) activity using in vitro assays to examine the growth of the *Escherichia coli* serotype, O101:H9, which is isolated from dairy calves and the Caco-2 cell line; it exhibited direct antibacterial effects, suppressed proinflammatory effects (such as IL-1β and IL-10 regulation), and reduced damage to the cell's integrity [178]. In vivo assays used on a specific pathogen-free (SPF) CD-1 neonatal mice model, significant antibacterial effects were also demonstrated, and they helped maintain colonic barrier integrity. In fact, UDCA supplementation attenuated colitis symptoms and recovered colonic short-chain fatty acid (SCFA) production. Through 16S rRNA gene sequencing, microbiotas from UDCA-treated neonatal mice ameliorated colitis symptoms, as evidenced by the successful colonization of bacteria, including *Oscillospiraceae*, *Ruminococcaceae*, *Lachnospiraceae*, and *Clostridia_UCG-014*, when compared with control and placebo microbiotas [178]. It is important to note that this prebiotic application was successful against an enteroaggregative *E. coli* (EAEC) and a multidrug-resistant extended-spectrum β-lactamase (ESBL)-producing *E. coli* isolate. Furthermore, probiotic applications are also evolving due to the isolation of new and more probiotic strains from a diverse set of samples, and more exhaustive in vitro and in vivo studies. For example, Kao and colleagues examined the extracellular electrons transferred from the honey-derived probiotic, *B. circulans*, which inhibits the human skin pathogen, *C. acnes*, by injecting the pathogen intradermally into mice ears to induce an inflammatory response [179]. The results showed that the in vitro *B. circulans* co-culture enhanced electron production and significantly suppressed *C. acnes* growth. Moreover, in the in vivo assays of the ears of the Crl:CD1(ICR) mice model, the *C. acnes* and macrophage inflammatory protein 2 (MIP-2) levels suggested that *B. circulans*-generated electrons affected *C. acnes* growth and alleviated the resultant inflammatory response [179]. Islam and colleagues also evaluated a new probiotic strain, *B. amyloliquefaciens* (BA PMC-80), which exhibited significant anti-*C. difficile* effects in a co-cultured in vitro assay [180]. It also exhibited no toxicity in a subchronic toxic in vivo hamster model; indeed, a reduction in infection severity and delayed death were observed. However, further studies are required to identify the antimicrobial compound produced by BA PMC-80, which would improve the treatment of the *C. difficile* infection (CDI) hamster model. Lastly, Ishnaiwer and colleagues found two new strains of *B. subtilis* (CH311 and S3B) and evaluated them against an ESBL-producing *E. coli* isolate, wherein both probiotic strains reduced ESBL-*E. coli*

titers by 4 log colony-forming units (CFU)/mL in an in vitro model of gut content [181]. However, an in vivo murine model of intestinal colonization showed no reduction in the fecal titers of the ESBL-*E. coli* strains, CH311 and S3B [181]. Thus, this study emphasizes the importance of in vivo experiments to identify efficient probiotics, and more importantly, to identify probiotic administration procedures that allow the development and improvement of effective delivery systems.

Other promising applications are synbiotics and postbiotics. Cui and colleagues applied *P. acidilactici* and phthalyl inulin nanoparticles (PINs) to be used against *S.* Gallinarum via an in vitro cocultivation assay [182]. The antibacterial activity of the symbiotic formulation was the highest among the treated groups (bacteria control and only PINs or probiotics), exhibiting a statistical reduction from log 8 to log 5 CFU/mL. Interestingly, PINs alone did not demonstrate any antibacterial activity, thus highlighting the synergistic inhibitory effect of this synbiotic formulation on this specific foodborne pathogen [182]. Moreover, Hashem and colleagues evaluated the encapsulation efficiency of alginate-$CaCl_2$ nanoparticles to be used against *S. cerevisiae*, as well as *Moringa oleifera* leaf extract (MOLE) to be used against multiple foodborne pathogens, including *E. coli* BA 12296B, *S. aureus* NCTC 10788, *C. albicans* ATCC MYA-2876), *L. monocytogenes* ATCC 19116, and *S.* Senftenberg ATCC 8400 [183]. The antimicrobial activity test for administered synbiotics uses the agar-well diffusion method, and it revealed significantly greater diameters for the inhibition zones of the nanoencapsulated synbiotic when compared with the nonencapsulated group against all tested pathogenic bacteria and fungi. Furthermore, in vivo assays used on the rabbit model produced no effects in terms of interleukin-l or immunoglobulin G (IgG) and IgA levels. Moreover, nanoencapsulated synbiotics significantly increased the number of beneficial intestinal and cecal microbes (yeast and lactic-acid bacteria) while reducing the number of coliforms and *Salmonella* sp. Lastly, in vitro gastrointestinal simulation tests revealed the highest protective effect for the survivability of the probiotic, *S. cerevisiae*, during gastric and intestinal enzymatic digestion [183]. In 2021, Hong and colleagues evaluated a new formulation of phthalyl pullulan nanoparticles (PPNs) to enhance the antimicrobial activity of the probiotic, *L. plantarum*, in a dysbiosis-induced murine model using the *Escherichia coli* K99 pathogen. The authors showed that the infection was significantly suppressed using synbiotics, and several well-known beneficial bacteria, such as *Lactobacillus* and *Bifidobacterium*, were enriched [167]. Likewise, in 2022, Bandyopadhyay and colleagues assessed the probiotic effect of two new *L. lactis* subsp. *lactis* strains against pathogenic and food spoilage bacteria and fungi (such as *Bacillus* sp., *E. faecalis*, *E. coli*, *L. monocytogenes*, *Aspergillus* sp., *C. albicans*, and *Fusarium oxysporum* among others); these probiotic strains demonstrated good antimicrobial activity against all pathogenic and food spoilage fungi tested in the study [168]. The further in vivo bacterial feeding of these strains for 30 days in Swiss albino mice either individually, or in combination with prebiotic inulin, improved gut colonization and immunoglobulin A (IgA) production levels. Additionally, the continued feeding provided health benefits that were better than the use of a commercial probiotic consortium together with a prebiotic mixture [168].

As previously stated, there are still not many in vivo studies on postbiotics. According to the National Center for Biotechnology Information (NCBI; https://www.ncbi.nlm.nih.gov/ accessed on 27 April 2023), most in vivo studies on postbiotics were published after 2019. Sornsenee and colleagues recently evaluated the in vitro antimicrobial effects, antioxidant activity, and anti-inflammatory effects of 10 lyophilized cell-free supernatants (LCFS) of *Lactobacillus* isolates from the fermented palm sap of trees from Southern Thailand, which were used against *E. coli* DMST4212, *A. baumannii* DMST 2271, *S. aureus* DMST 2928, and one clinical MRSA isolate [164]. All LCFS exhibited strong antibiofilm activity, they eradicated the biofilms formed by *A. baumannii* and *E. coli*, and they reduced the production of nitric oxide in RAW 264.7 cells [164]. Jung and colleagues analyzed the potential beneficial effects (for the host) of the LCFS mixture of *L. plantarum* KM1, *L. plantarum* KM2, and *B. velezensis* KMU01 (1:1:1; vol/vol) in RAW264.7 cells and a C57BL/6N mice model [184]; the study reported a significant reduction in tumor necrosis factor-alpha (TNF-$\alpha$) levels

and an increase in natural killer (NK) cell activation. In addition, the postbiotic mixture was able to modulate the abundance of Bifidobacteria, *Lachnospiraceae*, and *Lactobacillaceae* in the gut of the C57BL/6N mice model [184]. Golkar and colleagues also evaluated the immunomodulatory, anti-inflammatory, and antimicrobial activities of three individual postbiotic cold creams using the LCFS of *L. fermentum*, *L. reuteri*, and *B. subtilis* sp. *natto* to examine wound healing in a Sprague Dawley rat model [185]. Wound healing in animals using all three cold cream formulations exhibited faster recovery times when compared with animals that were given no treatment or only cold cream by itself. After day 4, all three postbiotic cold creams exhibited higher and significantly better wound healing abilities in comparison with the untreated group and the group treated with cold cream without postbiotics ($p < 0.0001$). The epithelialization process was complete in rats receiving *L. reuteri* and *B. subtilis* cold creams, whereas the *L. reuteri* cold cream inhibited the inflammation process in treated animals. Finally, animals treated with *L. reuteri* and *B. subtilis* cold creams did not demonstrate any histological alterations with regard to granulation [185]. Moreover, Puccetti and colleagues demonstrated the applicability of postbiotics via a spray-dried formulation of indole-3-carboxaldehyde (3-IAld) that was used against *Aspergillus fumigatus* using two cell lines (Beas-2B and Calu-3) and a C57BL/6 mice model in order to assay potential pulmonary toxicity and inflammatory cytokine gene expression, respectively [186]. The results demonstrated dual therapeutic benefits; the formulation can be used as an anti-inflammatory agent to prevent lung inflammation, and it can be used to reduce aspergillosis disease scores when locally delivered into the lungs via inhalable 3-IAld-Man powder [186].

These studies demonstrated the potential of synbiotics and postbiotics for immune system regulation and host-microbiome modulation. However, more in vitro, and especially in vivo, studies are needed to fully understand the impact and outcomes of synbiotics and postbiotics in host-microbiome and immune system responses against MDR pathogens before the post-antibiotic era begins.

## 6. Conclusions

The imminent 'post-antibiotic' era needs alternative therapies, innovation, and a multidisciplinary approach. The present review compiled the most recent studies that use different natural products to achieve this (more specifically, plant extracts, honey, propolis, prebiotics, probiotics, synbiotics, and postbiotics). These products, either alone or together with standard drug therapies, frequently enabled drug sensitivity in MDR pathogens to be restored, and treatments were improved via in vitro and in vivo assays. A new generation of non-antibiotic compounds is needed to fight MDR pathogens. Further studies evaluating these products, using genomics, transcriptomics, and proteomics, are necessary to address the gaps in the literature by identifying specific compounds and their potential to be used against MDR pathogens, thus allowing the development of new therapies.

**Author Contributions:** Conceptualization, A.M., F.A. and J.M.Á.-S.; methodology, A.M.; validation, A.M., F.A. and J.M.Á.-S.; formal analysis, A.M., F.A., L.Z.-M. and J.M.Á.-S.; investigation, A.M., F.A., L.Z.-M. and J.M.Á.-S.; resources, A.M.; data curation, A.M.; writing—original draft preparation, A.M., F.A., L.Z.-M. and J.M.Á.-S.; writing—review and editing, A.M., F.A. and J.M.Á.-S.; visualization, A.M.; supervision, A.M.; project administration, A.M.; funding acquisition, A.M. All authors have read and agreed to the published version of the manuscript.

**Funding:** This research was funded by the Collaboration Grants 2021 to António Machado of the Research Office from Universidad San Francisco de Quito USFQ, under Project ID: 17577 titled "The antibiofilm potential of lactobacilli biosurfactants against multi-drug-resistant pathogens". The funders had no role in the study design, data collection, analysis, decision to publish, or preparation of the manuscript.

**Institutional Review Board Statement:** Not applicable.

**Informed Consent Statement:** Not applicable.

**Data Availability Statement:** Not applicable.

**Acknowledgments:** A special recognition is due to all colleagues at the Microbiology Institute of USFQ, COCIBA, El Politécnico, and Research Office of Universidad San Francisco de Quito for their support in this study.

**Conflicts of Interest:** The authors declare no conflict of interest. The funders had no role in the design of the study; in the collection, analyses, or interpretation of data; in the writing of the manuscript; or in the decision to publish the results.

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
