# Peer review of "Use of Plant Extracts, Bee-Derived Products, and Probiotic-Related Applications to Fight Multidrug-Resistant Pathogens in the Post-Antibiotic Era"

_futurepharmacol, doi:10.3390/futurepharmacol3030034_

Round 1

Reviewer 1 Report

The manuscript is aimed to summarize the use of natural products to fight multidrug-resistant pathogens based on the title. However, the authors emphasized honey and propolis a lot in this manuscript, but the content about natural products is quite less. Therefore, the content of the manuscript doesnt really match the title. There are already some reviews on this topic, for example, Evidence-Based Complementary and Alternative Medicine. Volume 2021, Article ID 3663315; Natural Product Reports, DOI: 10.1039/b821648g; Frontiers in Microbiology 13 (2022).doi: 10.3389/fmicb.2022.887251. The authors did not discuss the difference between this review and those previous reports. These reviews are not cited yet. Therefore, a significant revision is required.

Author Response

Comments and Suggestions for Authors

The manuscript is aimed to summarize the use of natural products to fight multidrug-resistant pathogens based on the title. However, the authors emphasized honey and propolis a lot in this manuscript, but the content about “natural products” is quite less. Therefore, the content of the manuscript doesn’t really match the title.

Authors' answers: We agree with Reviewer 1’s comment, the present review combines three different research fields searching for alternative therapies against multidrug-resistant pathogens. As well-appointed by Reviewer 1, we rectified the title of the revised manuscript to represent the whole study and avoid misunderstanding of the Readers. We apologized for the mistake. The title is now changed as follow: “Use of plant extracts, bee-derived products, and probiotic-related applications to fight multidrug-resistant pathogens in the post-antibiotic era”.

There are already some reviews on this topic, for example, Evidence-Based Complementary and Alternative Medicine. Volume 2021, Article ID 3663315; Natural Product Reports, DOI: 10.1039/b821648g; Frontiers in Microbiology 13 (2022).doi: 10.3389/fmicb.2022.887251. The authors did not discuss the difference between this review and those previous reports. These reviews are not cited yet. Therefore, a significant revision is required.

Authors' answers: Thank you very much for your thoughtful suggestion. As well-suggested by Reviewer 1, we added the main findings obtained by these three reports in the subsection 2. Rise of MDR pathogens and future trends on the global infectious-disease crisis clarifying the difference between this review and those previous reports (Simões et al., 2009; Jubair et al., 2021; Chawla et al., 2022). Please check the new information on lines 97-101 and 134-170 of the revised manuscript with track-changes), more exactly:

Lines 97-101

“Although conjugation, transformation, and transduction are the three primary processes of HGT, six major types of mobile genetic elements (MGEs) have been characterized in the MDR pathogens such as transposons, gene cassettes and integrons, genomic islands, plasmids, bacteriophages, and integrative conjugative elements (ICEs) [24].”

Lines 134-170

“Despite the intrinsic resistance caused by the presence of EPS surrounding the microbial community within the biofilm, there is also several mechanisms of AMR previously described in other reports involving inactivation or modification of drug, limiting the drug uptake, drug target modifications, and decreasing the active concentration of drug inside the cell through drug efflux [24,33,34]. Physiological adaptation of microorganisms within the biofilm induces the development of intrinsic resistance and biofilms are the leading example of resistance to antimicrobial products. Biofilms have been reported to be 100–1000 times more resilient towards antimicrobials when compared to equivalent planktonic counterparts [33]. Besides the passage through EPS of the biofilm, the antimicrobial drug needs to enter the microorganism cell membrane at adequate concentration for a substantial time to perform its pharmacological action and produce its antimicrobial activity. The efflux pump mechanism is a common mechanism for numerous MDR pathogens to counter drugs extruding antimicrobial agents faster than usual [34]. Efflux pumps are proteinaceous and membranal transporters able to regulate the microbial cell cytoplasmatic environment and so remove toxins, antifungals, and antibiotics. Based on their sequence homologies, source of energy, substrate binding, and structural components, efflux transporters are usually classified into five prominent families [34], more exactly: resistance-nodulation division (RND); adenosine triphosphate (ATP)-binding cassette superfamily (ABC); multidrug and toxic compound extrusion (MATE); major facilitator superfamily (MFS); and small multidrug resistance family (SMR). The two main families of efflux pump proteins in fungi are the ABC and MFS transporters, while RND family is a specific group for Gram-negative bacteria. Finally, ABC, MATE, MFS, and SMR families are found to a large degree in Gram-positive and Gram-negative bacteria [34]. However, it is also known that different microorganisms can alter their membrane permeability through over- or under expression of porins, thus controlling the passage of several compounds through the cell membrane, which it has been widely reported among Gram-negative pathogens [34]. Finally, another well-known AMR mechanism is the degradation of antimicrobial agents or target site modifications through enzymes. In fact, different enzymes can remove or add a specific moiety to the antibiotic molecule or target site reaching a successful mutation on the pathogen against a certain drug [24]. The modification of antimicrobial drugs can be achieved through numerous biochemical reactions catalyzed by enzymes involving phosphorylation (e.g., chloramphenicol and aminoglycosides), acetylation (e.g., chloramphenicol, aminoglycosides, and streptogramins), and adenylation (e.g., lincosamides and aminoglycosides) [24]. Likewise, target site modifications of several drugs can be succeeded by point mutations in the gene encoding target site, enzymatic change of the target site, and bypassing the original site [24].”

References

  1. Chawla, M.; Verma, J.; Gupta, R.; Das, B. Antibiotic Potentiators Against Multidrug-Resistant Bacteria: Discovery, Development, and Clinical Relevance. Front Microbiol 2022, 13.
  2. Simões, M.; Bennett, R.N.; Rosa, E.A.S. Understanding Antimicrobial Activities of Phytochemicals against Multidrug Resistant Bacteria and Biofilms. Nat Prod Rep 2009, 26, 746–757.
  3. Jubair, N.; Rajagopal, M.; Chinnappan, S.; Abdullah, N.B.; Fatima, A. Review on the Antibacterial Mechanism of Plant-Derived Compounds against Multidrug-Resistant Bacteria (MDR). Evidence-based Complementary and Alternative Medicine 2021, 2021.

Author’s answer – We appreciated all suggestions and efforts made by Reviewer 1 that allowed us to improve the initial draft manuscript. We hope that Reviewer 1 find this version suitable for publication in the Future Pharmacology journal.

Reviewer 2 Report

Bacterial resistance is a worldwide problem. We should take a closer look at this problem and fight it at all costs, aiming at its complete elimination. The submitted manuscript deals with this topic, but some corrections should be made before publication. I would supplement the introduction with information on resistance in hospital conditions. What departments does it apply to and why? In order to make it more attractive and to increase the citation, a chapter on various derivatives, e.g. alkyl derivatives of naringenin (https://doi.org/10.3390/molecules25163642) should be added, among which the effect on highly resistant strains has been proven, including e.g. Helicobacter pylori. Throughout the Manuscript, the spelling of "in vitro" and "in vivo" should be corrected without italics. Please analyze the nomenclature of bacteria carefully. We write the full name if it appears for the first time. Abbreviation, if once again. There are omissions in this regard in the Manuscript (e.g. for S. cerevisiae, P. aeruginosa, but not only). Please review the entire Manuscript. "Clostridium difficile" - incorrect naming. Please change throughout the Manuscript to Clostridioides difficile. Salmonella names should also be corrected. If using for the first time, use the full name including both subsp. and serovar (in italics where applicable: https://www.ncbi.nlm.nih.gov/pmc/articles/PMC86943/). Only in the next use can we use only the name of the serovar.

Best regards

Author Response

Comments and Suggestions for Authors

Bacterial resistance is a worldwide problem. We should take a closer look at this problem and fight it at all costs, aiming at its complete elimination. The submitted manuscript deals with this topic, but some corrections should be made before publication.

Authors' answers: We want to thank Reviewer 2 for his/her constructive comments and thoughtful suggestions that allowed us to improve the original manuscript. We are pleased to inform you that thanks to your recommendations we have made the following changes:

I would supplement the introduction with information on resistance in hospital conditions. What departments does it apply to and why?

Authors' answers: As suggested by Reviewer 2, information on resistance in hospital conditions was added in the Introduction section, clarifying the most important antimicrobial resistance described in different infections and hospital sections.  Please check the new information on lines 47-64 of the revised manuscript with track-changes), more exactly:

Introduction section on lines 47-64

“Nowadays, a main public health concern is the health-care-associated infections (HAI). These infections are usually obtained onset 48 h after hospitalization although it may also occur after discharge of patients [10]. It is estimated that 7 and 10% of hospitalized patients in developing and developed countries acquired HAI [10,11], respectively. While around 3.2 million patients per year are affected by HAI in Europe [10]. The mortality rate and incidence among patients are normally correlated to the patients’ immunological status and geographical region. However, patients of burn units, intensive care units (ICUs), organ transplant receivers and neonates are the most common hospitalized groups affected by HAI [12,13]. In addition, these infections are also responsible for three out of four lethal cases in neonates in Sub-Saharan Africa and South-East Asia [10]. The most reported HAI are surgical site infections (2–5% incidence rate), catheter-related blood stream infections (12–25% incidence rate), catheter-related urinary tract infections (12% incidence rate), and ventilator-associated pneumonia (9–27% incidence rate) [14]. Currently, the most worrisome global AMRs are the plasmid-mediated spread of carbapenemases (e.g., KPC, NDM, VIM, OXA-48, and OXA-51) and colistin-resistance genes (mcr) in EnterobacteriaceaeAcinetobacter baumannii, and Pseudomonas aeruginosa, as well as vancomycin resistance gene (vanA) in Enterococcus sp. and Staphylococcus aureus, and methicillin resistance gene (mecA) in Saureus [10,14].”

References

  1. Avershina, E.; Shapovalova, V.; Shipulin, G. Fighting Antibiotic Resistance in Hospital-Acquired Infections: Current State and Emerging Technologies in Disease Prevention, Diagnostics and Therapy. Front Microbiol 2021, 12.
  2. Tacconelli, E.; Carrara, E.; Savoldi, A.; Harbarth, S.; Mendelson, M.; Monnet, D.L.; Pulcini, C.; Kahlmeter, G.; Kluytmans, J.; Carmeli, Y.; et al. Discovery, Research, and Development of New Antibiotics: The WHO Priority List of Antibiotic-Resistant Bacteria and Tuberculosis. Lancet Infect Dis 2018, 18, 318–327, doi:10.1016/S1473-3099(17)30753-3.
  3. Atiencia-Carrera, M.B.; Cabezas-Mera, F.S.; Tejera, E.; Machado, A. Prevalence of Biofilms in Candida Spp. Bloodstream Infections: A Meta-Analysis. PLoS One 2022, 17, e0263522, doi:10.1371/journal.pone.0263522.
  4. Cangui-Panchi, S.P.; Ñacato-Toapanta, A.L.; Enríquez-Martínez, L.J.; Reyes, J.; Garzon-Chavez, D.; Machado, A. Biofilm-Forming Microorganisms Causing Hospital-Acquired Infections from Intravenous Catheter: A Systematic Review. Curr Res Microb Sci 2022, 3, 100175, doi:10.1016/j.crmicr.2022.100175.
  5. Khan, H.A.; Baig, F.K.; Mehboob, R. Nosocomial Infections: Epidemiology, Prevention, Control and Surveillance. Asian Pac J Trop Biomed 2017, 7, 478–482.

In order to make it more attractive and to increase the citation, a chapter on various derivatives, e.g. alkyl derivatives of naringenin (https://doi.org/10.3390/molecules25163642) should be added, among which the effect on highly resistant strains has been proven, including e.g. Helicobacter pylori.

Authors' answers: As recommended by Reviewer 2, various derivatives (such as, terpenoids, polyphenols, and alkaloids alkyl derivatives of naringenin) from plant extracts and bee-derived products were added in the 3. Plant extracts and 4. Honey and propolis sections to increment the citations and value of the present review. Moreover, information of the reported effects by Duda-Madej et. al (2020) and other studies were described on highly resistant strains, including Helicobacter pylori and other well-known pathogens. However, we believe that the addition of a separate chapter or section on various derivates would distract the Readers on the main purpose of the present review, which aims to discuss three different research fields (plant extracts, bee-derived products, and probiotic-related applications) searching for alternative therapies against multidrug-resistant pathogens. However, it is important to mention that section 4 already described several derivatives of polyphenols and we only needed to add the importante of alkaloids in bee-derived products. Please check the new information on lines 199-243 at 3. Plant extracts section and on lines 339-357 at 4. Honey and propolis section of the revised manuscript with track-changes), more exactly:

  1. Plant extracts section lines 199-243

“Plant extracts are mainly constituted by two types of metabolites classified as primary and secondary compounds. Primary metabolites are essential compounds for plant survival, while secondary metabolites are usually formed in response to plant interaction with the environment [34]. Therefore, primary metabolites usually englobe glycolysis, shikimate pathway, and tricarboxylic acid cycle products among others, integrating nutrition and reproduction functions. However, these metabolites can also act as a precursor for thousands of secondary metabolites that are produced at different steps in primary metabolic pathways, producing new compounds that facilitate plant adaptation against environmental stress (e.g., bacteria, fungi, insect, disease, injury, temperature, and moisture) [34]. The molecules with great antimicrobial effects are usually secondary metabolites, where the molecules identified in most studies are terpenoids, polyphenols (such as flavonoids, stilbenes, lignans, and phenolic acids), and alkaloids. Many plant terpenoids have found fortuitous uses in medicine [48]and their antimicrobial activity has been attributed to their general membrane disrupting properties [48]. For example, terpenoids of Syzigium cumini leaves evidenced antibacterial activity against MRSA and pathogenic E. coli by minimum inhibitory concentration (MIC) and minimum bactericidal concentration (MBC) assays [34]. Likewise, polyphenols’ antimicrobial effects were also documented [34], besides their properties as an antioxidant, anti-inflammatory, anticancer, and antihypertensive activities [34,49]. Although the exact mechanism for polyphenols’ antimicrobial action is not fully understood, several polyphenols were reported with antimicrobial activity against MDR pathogens. Studies postulated different mechanisms at the cellular level, where polyphenols can bind to bacterial enzymes via a hydrogen bond, inducing several modifications in cell membrane permeability and cell wall integrity [34,49]. Numerous reports were focused on the most abundant flavonoids, such as flavanols (e.g., quercetin and kaempferol), evidencing potent antimicrobial activity against Gram-positive and Gram-negative pathogens, as well as resistant strains [34]. As previously described, the combination of quercetin with amoxicillin exhibited synergistic activity against amoxicillin-resistant Staphylococcus epidermidis isolates [49]. Bryophyllum pinnatum extract revealed kaempferol and derivatives with significant antimicrobial activity against several bacterial and fungi pathogens, including antibiotic-resistant S. aureus and P. aeruginosa as well as Candida species and Cryptococcus neoformans [50]. Kaempferol-mediated inhibition of NorA efflux pump was postulated as its action’s mechanism against S. aureus [51]. Finally, alkaloids are organic nitrogenous compounds with structural diversity and their antimicrobial activity has been reported since the 1940s [34]. The mechanism of alkaloids against various microbial pathogens is characterized by efflux pump inhibition [52]. In 2020, Duda-Madej and colleagues demonstrated the antibacterial activity of 18 compounds of O-alkyl derivatives of naringenin and their oximes against clinical isolates of clarithromycin-resistant Helicobacter pylori, vancomycin-resistant Enterococcus faecalis, methicillin-resistant Staphylococcus aureus, and beta-lactam-resistant Acinetobacter baumannii and Klebsiella pneumoniae. [53]. From the pathogen group set, the clarithromycin-resistant strain of H. pylori showed the highest susceptibility to most of the 18 compounds. Moreover, when evaluating the synergy between O-alkyl derivatives/oximes and several antibiotics by fractional inhibitory concentration index (FICI), a potent synergy was observed against H. pylori, S. aureus, and E. faecalis [53].”

  1. Honey and propolis section on lines 339-357

“Phenols, flavonoids, terpenes, and alkaloids are also included in the group of antimicrobial-related compounds [84], where flavanols are one of the most abundant flavonoids present in food (such as honey and propolis). Flavanols are well-known for their potent antimicrobial activity against Gram-positive and Gram-negative pathogens, including resistant strains[34]. Meanwhile, little is stil known about the types of alkaloid compounds in the floral origin of numerous honey products. However, in 2021, Jaktaji and Ghalamfarsa evaluated the interaction of three monofloral honeys (Avishan, Gavan, and Konar) with ciprofloxacin against E. coli [85]. This study demonstrated that all three honeys-ciprofloxacin combinations decreased the viability of MG1655 and M1 E. coli strains more than ciprofloxacin alone. Moreover, the combination of these honeys and alkaloid extract of Sophora alopecuroides enhanced the anti-pump activity and reduced the oxidative stress response of the E. coli. Recently, Jaktaji and Koochaki evaluated the in vitro activity of honey and alkaloid extract of Sophora alopecuroides in combination with antibiotics against four biofilm‐producing Paeruginosa isolates [86]. This study revealed the synergistic effect of alkaloid extract‐honey in combination with ciprofloxacin against all Paeruginosa isolates, showing a significantly decrease of the antibiotic resistance and expression of the mexA gene [86]. Both studies demonstrated the importance of alkaloids from plant extracts and honeys as a source of antimicrobial agents and their combination with standard drugs against MDR and biofilm-associated pathogens [85,86].”

References

  1. Jubair, N.; Rajagopal, M.; Chinnappan, S.; Abdullah, N.B.; Fatima, A. Review on the Antibacterial Mechanism of Plant-Derived Compounds against Multidrug-Resistant Bacteria (MDR). Evidence-based Complementary and Alternative Medicine 2021, 2021.
  2. Bergman, M.E.; Davis, B.; Phillips, M.A. Medically Useful Plant Terpenoids: Biosynthesis, Occurrence, and Mechanism of Action. Molecules 2019, 24.
  3. Daglia, M. Polyphenols as Antimicrobial Agents. Curr Opin Biotechnol 2012, 23, 174–181.
  4. Tatsimo, S.J.N.; Tamokou, J. de D.; Havyarimana, L.; Csupor, D.; Forgo, P.; Hohmann, J.; Kuiate, J.-R.; Tane, P. Antimicrobial and Antioxidant Activity of Kaempferol Rhamnoside Derivatives from Bryophyllum Pinnatum. BMC Res Notes 2012, 5, 158, doi:10.1186/1756-0500-5-158.
  5. Holler, J.G.; Christensen, S.B.; Slotved, H.-C.; Rasmussen, H.B.; Guzman, A.; Olsen, C.-E.; Petersen, B.; Molgaard, P. Novel Inhibitory Activity of the Staphylococcus Aureus NorA Efflux Pump by a Kaempferol Rhamnoside Isolated from Persea Lingue Nees. Journal of Antimicrobial Chemotherapy 2012, 67, 1138–1144, doi:10.1093/jac/dks005.
  6. Khameneh, B.; Iranshahy, M.; Soheili, V.; Fazly Bazzaz, B.S. Review on Plant Antimicrobials: A Mechanistic Viewpoint. Antimicrob Resist Infect Control 2019, 8, 118, doi:10.1186/s13756-019-0559-6.
  7. Duda-Madej, A.; Kozłowska, J.; Krzyżek, P.; Anioł, M.; Seniuk, A.; Jermakow, K.; Dworniczek, E. Antimicrobial O-Alkyl Derivatives of Naringenin and Their Oximes Against Multidrug-Resistant Bacteria. Molecules 2020, 25, 3642, doi:10.3390/molecules25163642.
  8. Brudzynski, K. Honey as an Ecological Reservoir of Antibacterial Compounds Produced by Antagonistic Microbial Interactions in Plant Nectars, Honey and Honey Bee. Antibiotics 2021, 10, 551, doi:10.3390/antibiotics10050551.
  9. Jaktaji, R.P.; Ghalamfarsa, F. Antibacterial Activity of Honeys and Potential Synergism of Honeys with Antibiotics and Alkaloid Extract of Sophora Alopecuroides Plant against Antibiotic-Resistant Escherichia Coli Mutant. Iran J Basic Med Sci 2021, 24, 623–628, doi:10.22038/IJBMS.2021.54224.12179.
  10. Jaktaji, R.P.; Koochaki, S. In Vitro Activity of Honey, Total Alkaloids of Sophora Alopecuroides and Matrine Alone and in Combination with Antibiotics against Multidrug-Resistant Pseudomonas Aeruginosa Isolates. Lett Appl Microbiol 2022, 75, 70–80, doi:10.1111/lam.13705.

Throughout the Manuscript, the spelling of "in vitro" and "in vivo" should be corrected without italics. Please analyze the nomenclature of bacteria carefully. We write the full name if it appears for the first time. Abbreviation, if once again. There are omissions in this regard in the Manuscript (e.g. for S. cerevisiae, P. aeruginosa, but not only). Please review the entire Manuscript.

"Clostridium difficile" - incorrect naming. Please change throughout the Manuscript to Clostridioides difficile. Salmonella names should also be corrected. If using for the first time, use the full name including both subsp. and serovar (in italics where applicable: https://www.ncbi.nlm.nih.gov/pmc/articles/PMC86943/). Only in the next use can we use only the name of the serovar.

Authors' answers: As well-detected by Reviewer 2, the term “in vivo” and “in vitro” were changed in the non-italics form. The nomenclature of microorganims’ names was carefully revised, in particular Salmonella names using at first the full name including both subsp. and serovar, and then we only stated the name of the serovar as suggested by Reviewer 2. Finally, we replaced “Clostridium difficile” with the correct name “Clostridioides difficile”. We apologized for these mistakes, and we rectified these errors in all text of the manuscript to avoid misunderstanding of the Readers (Please check these amendments on the main text and tables of the revised manuscript with track-changes).

Best regards

Author’s answer – We appreciated all suggestions and efforts made by Reviewer 2 that allowed us to improve the initial draft manuscript. We hope that Reviewer 2 find this version suitable for publication in the Future Pharmacology journal.

Round 2

Reviewer 1 Report

All comments are properly addressed.

Reviewer 2 Report

Dear Authors,

Thank you for the corrections made. Those that were necessary were accepted by the Authors. I am satisfied. I wish the Authors good luck and many more interesting works like this one.